# Intermediate-complexity Parameterisation of Blowing Snow in the ICOLMDZ AGCM: development and first applications in Antarctica

Étienne Vignon[*,1,4], Nicolas Chiabrando[*,1], Cécile Agosta[2], Charles Amory[3], Valentin Wiener[1], Justine Charrel[1], Thomas Dubos[1], and Christophe Genthon[1]

[*]These authors contributed equally to this work.
[1]Laboratoire de Météorologie Dynamique-IPSL, Sorbonne Université/CNRS/ Ecole Normale Supérieure-PSL Université/ Ecole Polytechnique-Institut Polytechnique de Paris, Paris, France
[2]Laboratoire des Sciences du Climat et de l'Environnement, LSCE/IPSL, CEA-CNRS-UVSQ, Université Paris-Saclay, 91191 Gif-sur-Yvette, France
[3]Univ. Grenoble Alpes/CNRS/IRD/G-INP/INRAE, Institut des Geosciences de l'Environnement, Grenoble, France
[4]Laboratoire de Physique et Chimie de l'Environnement et de l'Espace (LPC2E), Université d'Orléans, CNRS UMR7328, CNES, Orléans, France

**Correspondence:** Étienne Vignon (etienne.vignon@lmd.ipsl.fr)

**Abstract.** Recent regional model findings suggest that the aeolian erosion of surface snow is a significant contribution to the overall Antarctic surface mass balance (SMB) through ice crystals sublimation and export outside of the ice sheet. Such findings raise the question of the relevance of accounting for such a process also in global climate models. This study presents the development of an intermediate-complexity parameterisation of blowing snow for the ICOLMDZ atmospheric general circulation model, the atmospheric component of the IPSL Coupled Model. The parameterisation is designed to be a trade-off between physical complexity and applicability in a general circulation model, with constraints on numerical cost and stability. The parameterisation is evaluated with in situ observations using limited-area simulations over Adélie Land. The model exhibits satisfactory results in terms of summer wind speed, temperature and intensity of blowing snow fluxes. In winter, blowing snow intensity and occurrences are overestimated close to the coast, concurring with a positive wind speed bias. In terms of blowing snow occurrences throughout the year, ICOLMDZ exhibits comparable performance with the regional atmospheric model MAR. Boundary-layer moistening and cooling as well as changes in surface radiative fluxes due to blowing snow crystals are also quantified in the simulations. Global simulations at standard global climate model resolution are carried out to investigate how the Antarctic SMB is modified with the activation of the blowing snow parameterisation. Results show an overall decrease of the net snow accumulation in the escarpment region due to surface snow erosion and an increase along the coast due to blowing snow deposition and increase in precipitation.

## 1 Introduction

The aeolian erosion of surface snow is an important component of the atmospheric branch of the Antarctic water cycle (Frezzotti et al., 2004). The snow mass sublimated during transport by the wind as well as its export out of the continent

are net losses from the point of view of the ice sheet. Aeolian snow erosion, transport and deposition (processes commonly referred to as drifting and blowing snow) have been shown to significantly affect the surface mass balance (SMB) of the Antarctic at the local scale (e.g., (Lenaerts et al., 2012a; Amory et al., 2021)), especially in coastal and escarpment regions where strong katabatic winds develop, leading to an intense export and sublimation of airborne snow (e.g. (Scarchilli et al., 2010; Palm et al., 2017)). Subsequently drifting and blowing snow have been parameterised in a few meso-scale and regional atmospheric models mostly for local to continental studies (e.g., Lenaerts et al., 2012b; Vionnet et al., 2014; Gallée et al., 2001; Gerber et al., 2023).

However, the effects of drifting and blowing snow on the overall Antarctic ice sheet climate and SMB are still debated. This particularly questions to what extent a parameterisation of those processes in global climate models is relevant and justified. Hereafter, we will combine blowing and drifting snow into the single denomination of blowing snow for convenience.

Le Toumelin et al. (2021) reveal significant effect of blowing snow on the surface radiative and turbulent fluxes over coastal Antarctica which suggests the possible importance of such a process for the surface energy budget over the ice sheet margins, a region particularly critical for global climate due to the melting and destabilisation of ice-shelves as well as intense atmosphere - sea ice - ocean interactions. Moreover, continental-scale regional simulations with the CRYOWRF model in Gerber et al. (2023) suggest that 4.2% of the annual Antarctic precipitation is removed by drifting and blowing snow among which 1% through direct export off the continent. This 4.2% estimate is quite similar to previous estimates using the RACMO model (Lenaerts and van den Broeke, 2012), suggesting that blowing snow significantly influences the SMB of the whole Antarctic ice sheet through export and sublimation (Gadde and van de Berg, 2024). In addition, blowing snow has been shown to affect the formation and structure of clouds in polar regions when it results from the aeolian erosion of snow above sea-ice that contains a significant amount of sea-salt. When blowing snow crystals sublimate in the atmosphere, sea-salt aerosols are released thereby increasing the amount of cloud condensation nuclei and influencing cloud formation and microphysical properties (Yang et al., 2019; Gong et al., 2023).

Such elements are strong motivations for assessing the effects of including blowing snow in a global climate model. Several parameterizations of snow erosion and transport have been proposed so far (e.g., Gallée et al., 2001; Lenaerts et al., 2012b; Vionnet et al., 2014; Sharma et al., 2023). However, to our knowledge, all of them were developed for mesoscale models and often involve a level of complexity — as well as an additional computational cost, particularly due to the inclusion of extra water species — that is not always compatible with the constraints of global climate simulations. Moreover their applicability with typical vertical grids and time steps used in global models has not been assessed and questions regarding numerical integration aspects and validity of turbulent mixing formulations can emerge.

The present paper presents the development and tests of an intermediate-complexity parameterisation of blowing snow for the ICOLMDZ atmospheric general circulation model (AGCM). ICOLMDZ is currently being developed for carrying out future projections of the Antarctic water cycle and past SMB reconstructions in the framework of the AWACA project (https://cordis.europa.eu/project/id/951596) and including a blowing snow parameterisation has been identified as a development priority.

The paper is structured as follows. Section 2 presents the design of the parameterisation and its integration into the ICOLMDZ model. Section 3 then presents two examples of application in regional simulations over Adélie Land and in global simulations with a particular focus on the impact of simulated blowing snow on the Antarctic SMB. Section 4 closes the paper with discussions and conclusions.

## 2 Blowing snow parameterisation in ICOLMDZ

### 2.1 Preamble: the ICOLMDZ AGCM and its application for polar research

The ICOLMDZ AGCM consists in the coupling of the DYNAMICO icosahedral dynamical core (Dubos et al., 2015) and the physics of the LMDZ AGCM (Hourdin et al., 2020), the atmospheric component of the IPSL-CM global climate model (Boucher et al., 2020). LMDZ has been used for several Antarctic studies, in particular for works on the Antarctic SMB (e.g., Agosta et al., 2013), for investigations on the oceanic forcing on the Antarctic climate (Krinner et al., 2014), for analyses of the boundary layer on the Plateau (Vignon et al., 2018) as well as for works on precipitation on the Antarctic coast (Roussel et al., 2023), and stable water isotopes (Cauquoin et al., 2019; Dutrievoz et al., 2025).

Even though some work is underway to improve the representation of the surface snow over ice sheet surfaces in the ORCHIDEE model (Charbit et al., 2024), the land-surface component of the IPSL Earth System Model coupled with ICOLMDZ (Cheruy et al., 2020; Arjdal et al., 2024), the exchanges of energy and water between the atmosphere and so-called 'land-ice' surfaces - encompassing both the Greenland and Antarctic ice-sheets - are still treated by a separate simple snow scheme in the LMDZ model (Vignon et al., 2017; Le Moigne et al., 2022). This quite crude snow scheme assumes constant values for the visible and near-infrared broadband albedos, constant values for the momentum and thermal roughness lengths and the heat transfer in the snow is parameterised as a conductive process with a fixed thermal inertia whose value has been fixed to that of typical snow found on the high Antarctic Plateau. Surface snow density is not a variable of the scheme. Melting is parameterised as a bulk process and the melt water is directly transferred to the ocean. The refreezing of liquid water in the snowpack is not taken into account.

In this study, we consider the version of the LMDZ physics package currently in development for the 7th exercise of the Coupled Model Intercomparison Project (CMIP7). It is mostly based on that used for CMIP6 (Hourdin et al., 2020; Madeleine et al., 2020) but we employ the new TKE-l turbulent diffusion scheme developed in Vignon et al. (2024) that exhibits better numerical properties as well as more robust and more easily tunable formulations of the different terms of the eddy diffusivity coefficients compared to the previous TKE-l scheme of the model (Vignon et al., 2017). Moreover, this new scheme considers a turbulent mixing length formulation that depends on the wind shear in stable conditions following Grisogono and Belušić (2008) which is particularly important in flows with strong wind shear such as Antarctic katabatic jets. Wiener et al. (2025) recently conducted an extensive assessment of the ability of ICOLMDZ to simulate katabatic winds along the Antarctic slopes with this specific model configuration. They show that the model is able to reliably simulate the surface winds but also raise the need for further development regarding the parameterisation of the snow surface roughness and albedo to better capture

the spatio-temporal variability of the wind. Concurring with previous studies (e.g., Gallée et al., 2013; Vignon et al., 2019), Wiener et al. (2025) also underline the difficulty to capture the correct location and magnitude of the coastal transition of the katabatic layer through a so-called 'katabatic jump', which manifests as sudden decrease in surface wind speed in a few km.

## 2.2 General concepts of the blowing snow parameterisation

As ICOLMDZ is primarily the atmospheric component of a global climate model and not a meso-scale model developed for fine-scale studies in complex terrain areas, the question of the degree of sophistication required for a new blowing snow parameterisation must be raised. The answer of course depends on the objectives and on the desired applications and also, on the existing structure of the model namely the typical horizontal and vertical resolutions at which it is run and its physical package. Here, we aim to equip ICOLMDZ with a blowing snow scheme to capture the main snow transport events that can substantially affect the Antarctic SMB and potentially the polar hydrological cycle at regional and continental scales.

We therefore follow an intermediate-complexity approach in the sense that the parameterisation does not require a very sophisticated snow scheme - such as SNOWPACK for CRYOWRF for instance (Sharma et al., 2023) - and does not include an additional discretization of the surface layer as in Vionnet et al. (2014). Such as in MAR (Gallée et al., 2001), RACMO (Lenaerts et al., 2012b) and WRF (Saigger et al., 2024), a blowing snow flux is directly calculated between a fully parameterised saltation layer near the surface and the first model level at a few meters above the ground surface. However, the specific content of blowing snow particles in suspension $q_b$ (in $\mathrm{kg\,kg^{-1}}$) is treated as an independent water variable in the model - unlike in MAR for instance - to properly distinguish the blowing snow contribution to precipitation and radiative effects from that of typical clouds. $q_b$ is advected by the dynamical core and vertically transported by turbulent diffusion and sedimentation. More specifically, $q_b$ obeys the following evolution equation:

$$\frac{\partial q_b}{\partial t} = \frac{\partial q_b}{\partial t}\bigg|_{adv} + \frac{\partial q_b}{\partial t}\bigg|_{turb} + \frac{\partial q_b}{\partial t}\bigg|_{sub} + \frac{\partial q_b}{\partial t}\bigg|_{melt} + \frac{\partial q_b}{\partial t}\bigg|_{sed} \tag{1}$$

where the subscript $adv$ refers to the advection by the dynamical core and the subscripts $sub$, $sed$, $melt$, $turb$ to the parameterized sublimation, sedimentation, melting and turbulent diffusion processes respectively. Nonetheless, note that we keep a one-moment treatment for the blowing snow water species and do not consider an additional prognostic estimation of the number of blowing snow particles (Vionnet et al., 2014; Sharma et al., 2023).

## 2.3 Surface snow erosion

The first part of the new blowing snow scheme is a parameterisation of surface snow erosion following Gallée et al. (2001) and Amory et al. (2021). It consists of calculating a blowing snow flux from a fully parameterised saltation layer near the surface to the first model level with a drag coefficient that is directly calculated from atmospheric variables at the first model level. Snow erosion is calculated only over land-ice surfaces and therefore concerns only the Greenland and Antarctic ice sheets in global simulations. Although we acknowledge the added value of additional vertical discretisation of the surface layer to better capture the sharp gradients of blowing snow near the ground surface (Vionnet et al., 2014; Sharma et al., 2023), we choose a

simpler framework here to keep the standard vertical grid of the model and because we mostly aim to simulate the main aeolian snow transport events during which the blowing snow is well mixed over the first meters of the atmosphere.

Following Gallée et al. (2001) and Amory et al. (2021), we assume that blowing snow particles are ejected from the saltation
layer when the friction velocity $u_*$ exceeds a threshold value $u_{*,t}$ that reads:

$$u_{*,t} = u_{*,t0} e^{\left(\frac{\rho_i}{\rho_{s,0}} - \frac{\rho_i}{\rho_s}\right)} e^{\max(0,\rho_s - \rho_{s,\infty})} \tag{2}$$

where $\rho_i = 917 \text{ kg m}^{-3}$, and $\rho_{s,0} = 300 \text{ kg m}^{-3}$ are two fixed parameters corresponding to the density of ice and fresh snow respectively. $u_{*,t0}$ is the so-called standard threshold friction velocity expressed following Gallée et al. (2001):

$$u_{*,t0} = \frac{\log 2.688 - \log 1 + 0.75 d_s - 0.5 s_s + 0.5}{0.085} C_D^{0.5} \tag{3}$$

where $C_D$ is the drag coefficient for momentum. $s_s$ and $d_s$ are the sphericity and dendricity of snow grains set to 0.5 as in Amory et al. (2021) to reduce the number of sensitivity parameters. Note that the rightmost exponential term in equation 2 has been added here to limit the erosion to occur when the surface snow density $\rho_s$ approaches $\rho_{s,\infty} = 450 \text{ kg m}^{-3}$, as in Amory et al. (2021). It is worth recalling that the surface snow density $\rho_s$ is not a variable of the surface scheme over land-ice surfaces in the model. Therefore, we have to provide an estimate of $\rho_s$ to properly compute the erosion threshold. For this purpose,
while LMDZ is not coupled to an advanced snow scheme over ice sheets, we propose a relatively simple heuristic approach.

If snow precipitation – excluding sedimentation of blowing snow – has occurred during a given time step, the snow density is assumed to be that of fresh snow $\rho_{s,0}$. If all the snowfall accumulated during the time step has been eroded, we consider the erosion of the underlying snow layer whose density value $\rho_s$ is determined with a simple model of densification with snow age:

$$\rho_s = \rho_{s,0} + (\rho_{s,\infty} - \rho_{s,0})(1 - e^{-a_s/\tau_d}) \tag{4}$$

where $a_s$ is the snow age and $\tau_d$ is a snow densification time scale. Within each time step $\Delta t$, we do not *a priori* know the time that corresponds to the erosion of the superficial fresh surface snow – which is the snow that has fallen during the time step – and the time that corresponds to the erosion of the underlying, and thus older, snow layers. We thus assume that the fresh snow erosion occurs during a fraction $\omega_f$ of $\Delta t$ that depends on the relative difference between the fresh snow erosion flux $Er$ and
the snowfall during the time step $Sf$: $\omega_f = e^{-\left(\frac{|Er-Sf|}{Sf}\right)}$ The snow age is reset to 0 as soon as some fresh snow accumulates during the time step that is, if some fresh snow corresponding to the snow that falls at the given time step remains after the erosion process.

To account for the negative feedback of snow erosion on snow density (Amory et al., 2016, 2017) as well as the effect of rainfall on density (Marshall et al., 1999), we propose a simple heuristic expression for the surface snow densification time
scale $\tau_d$:

$$\tau_d = \max(\tau_{d,min}, \tau_{d0} e^{\left(-\frac{P_{bs}}{P_{bs,t}} - \frac{P_r}{P_{r,t}}\right)} e^{\left(-\max\left(\frac{T_s - T_0}{\Delta T_0}, 0.\right)\right)}) \tag{5}$$

where $\tau_{d,0}$ is the densification time scale in absence of snow erosion, rain and melting. It has been set to 10 days following careful inspection of the evolution of the snow density in MAR simulations over the Antarctic (not shown). $\tau_{d,min}$ is the

densification time scale in presence of very intense snow ablation or rain. It has been set to 1 day, which correspond to the rain-induced snow densification time scale according to Marshall et al. (1999) and to the average duration of drifting-snow events - and for exhaustion of erodible snow to be reached - according to Antarctic observations in Amory (2020). $P_{bs}$ (resp. $P_r$) is the sedimentation flux of blowing snow (resp. rainfall flux) at the surface and $P_{bs,t}$ (resp $P_{r,t}$) a threshold value set to $0.01 \ \mathrm{kg \ m^{-2} \ s^{-1}}$. The rightmost term accounts for the sharp decrease in $\tau_d$ during snow melting, $T_s$ being the snow surface temperature, $T_0 = 273.15 \ \mathrm{K}$ and $\Delta T_0 = 1 \ \mathrm{K}$.

The depth of the saltation layer is calculated following Pomeroy (1989):

$$h_{salt} = 0.08436 u_*^{1.27} \tag{6}$$

The concentration of aeolian snow at the top of the saltation layer - i.e. the lower boundary condition for $q_b$ - is estimated using steady-state and vertically-homogeneous model of saltation layer of Pomeroy (1989) as in Gallée et al. (2001):

$$q_{b,salt} = \frac{e_{salt}}{g h_{salt}} (u_*^2 - u_{*,t}^2) \tag{7}$$

where $e_{salt} = (3.25 u_*)^{-1}$ is the saltation efficiency. It is worth mentioning that the parameterisation of saltation for large-scale models is an active area of research (Melo et al., 2024) and we leave the assessment of the $q_{b,salt}$ formulation sensitivity for future studies.

The vertical blowing snow flux from the surface towards the atmosphere $\left. \overline{\rho w' q_b'} \right|_s$ then reads:

$$\left. \overline{\rho w' q_b'} \right|_s = -\rho u_* q_{b*} = \max(-\rho C_{Db} U (q_b - q_{b,salt}), \mathrm{F_{max}}) \tag{8}$$

where $\rho$ is the air density, $U$ the wind speed at the first model level, $q_{b*}$ is the turbulent scale of $q_b$ and $F_{max}$ is a higher-bound for snow erosion. The latter is calculated such that all the snow in the saltation layer cannot be removed during one single time step (and is therefore time-step dependent). We take the drag coefficient for blowing snow $C_{Db}$ equal to that for heat and water vapor. In presence of drifting or blowing snow, the Monin-Obukhov similarity theory - on which are based the surface turbulent bulk flux formulae used in models - fails in correctly predicting the turbulent fluxes of sensible and latent heat. In fact, exchanges of heat and moisture associated with aeolian snow particles sublimation make the assumption of height-constant turbulent fluxes in the surface layer no longer valid. This leads to strong underestimations of sensible and latent heat exchanges (Sigmund et al., 2022). To the authors' knowledge, there is currently no reliable formula for the turbulent drag coefficients for heat, moisture and blowing snow in presence of aeolian snow transport in the surface layer, especially for application in models with a first atmospheric level at a few meters above the ground surface. We leave this aspect for further research.

## 2.4 Turbulent transport

The specific content of blowing snow is vertically mixed by the TKE-l turbulent diffusion scheme of LMDZ through the resolution of the diffusion equation:

$$\left. \frac{\partial q_b}{\partial t} \right|_{turb} = -\frac{1}{\rho} \frac{\partial \overline{\rho w' q_b'}}{\partial z} = \frac{1}{\rho} \frac{\partial}{\partial z} (\rho K_b \frac{\partial}{\partial z} q_b) \tag{9}$$

Once the $K_b$ eddy diffusion coefficient has been calculated at vertical model layer interfaces, such an equation is numerically solved with an implicit approach through the inversion of a tri-diagonal matrix. $K_b$ is taken proportional to that for momentum $K_m$ i.e.:

$$K_b = \zeta_b K_m \tag{10}$$

There is a lack of clarity in the literature about the values of $\zeta_b$. While Déry and Yau (2001) sets $\zeta_b = 1$ in their blowing snow simulation, observations of Mann (1998) suggest $\zeta_b$ values greater than unity. Amory et al. (2021) emphasise that such a parameter can be tuned to compensate for a likely overestimation or underestimation of the settling velocity of blowing-snow particles. In the present study, we set $\zeta_b = 1$ and will preferentially adjust the settling velocity defined hereafter.

It is worth noting here that we neglect the effect of blowing snow on local stratification in the buoyancy production of TKE (Gallée et al., 2001) as its contribution to the overall TKE budget and its impact on the overall TKE profile are generally small above the first meter above the ground (Bintanja, 2000).

## 2.5 Sublimation, melting and sedimentation

The parameterisation of blowing snow sublimation is inspired by that commonly used for cloud ice crystals detailed in Pruppacher et al. (1998). We assume that the blowing snow particles population obey a monodispersed distribution of spherical ice crystals of density $\rho_b$ and radius $r_b$ that is set to $50~\mu$m by default. The height-dependent radius formulation of Saigger et al. (2024) has also been implemented but not fully tested yet. The loss of $q_b$ due to sublimation then reads (Rutledge and Hobbs, 1983; Muench and Lohmann, 2020):

$$\begin{aligned}
\left.\frac{\partial q_b}{\partial t}\right|_{sub} &= -\left.\frac{\partial q_v}{\partial t}\right|_{sub} \\
&= -\gamma_{sub}\frac{6\rho}{\rho_b \pi r_b^2 (A' + B')}\left(1 - \frac{q_v}{q_{si}}\right)q_b
\end{aligned} \tag{11}$$

where $q_v$ is the specific humidity of the air, $q_{si}$ the saturation specific humidity with respect to ice, $A'$ and $B'$ two thermodynamic functions of temperature whose detailed expressions are given in Pruppacher et al. (1998). $\gamma_{sub}$ is a tuning coefficient that controls the intensity of the sublimation process and whose default value has been set to 0.01 after preliminary comparisons of observed and simulated near-surface relative humidity fields (not shown). The sublimation rate is limited to prevent the specific humidity to exceed saturation with respect to ice. The effect of blowing snow sublimation on the evolution of temperature and water vapour is taken into account. It is worth noting that during strong blowing snow events, significant amount of blowing snow can enter a relatively dry layer leading to intense and abrupt sublimation which can be quite challenging to resolve in time with the typical coarse time steps used in AGCMs. In fact, both $q_v$ and $q_b$ can substantially vary during a time step $\Delta t$ and given that the sublimation rate depends on the two variables, the numerical resolution of Eq. (11) is a highly relevant issue for a blowing snow parameterisation in an AGCM. We propose here a 'double implicit' numerical treatment for both $q_b$ and $q_v$ that is Eq. (11) then reads:

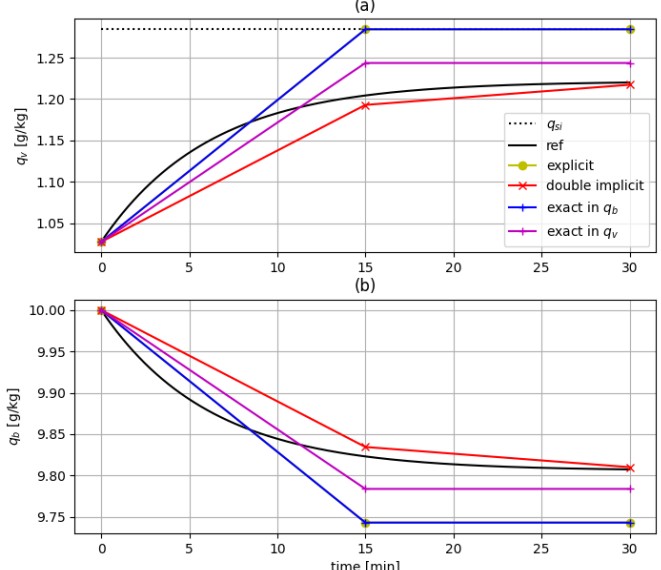

**Figure 1.** Idealised blowing snow sublimation experiment with a numerical toy model of Eq. 11 with different numerical methods (details in the main text of Sect. 2.5 ). Initial conditions are temperature $T = 260$ K, pressure $P = 95000$ Pa, relative humidity wrt ice $RH_i = 80$ %, $q_b = 10$ g kg$^{-1}$. The time step used is 15 min. Panel a (resp. b) shows the evolution of $q_v$ (resp. $q_b$). The solid black lines show the reference solution obtained with a 1 s time step (for which all methods converge). In panel a, the dotted black line shows the saturation value with respect to ice. Note that the blue and yellow curves are so close that they look superimposed.

$$\left. \frac{q_b^{t+\Delta t} - q_b^t}{\Delta t} \right|_{sub} = - \left. \frac{q_v^{t+\Delta t} - q_v^t}{\Delta t} \right|_{sub} \tag{12}$$

$$= -\gamma_{sub} \underbrace{\frac{3}{\rho_b r_b^2 (A' + B')}}_{\xi} (1 - \frac{q_v^{t+\Delta t}}{q_{si}^t}) q_b^{t+\Delta t} \tag{13}$$

which after some rearrangement can read:

$$\gamma_{sub}\xi\frac{\Delta t}{q_{si}^t}(q_b^{t+\Delta t})^2 + \left(1 + \gamma_{sub}\xi\Delta t - \gamma_{sub}\xi\frac{q_b^t\Delta t}{q_{si}^t} - \gamma_{sub}\xi\frac{q_v^t\Delta t}{q_{si}^t}\right)q_b^{t+\Delta t} - q_b^t = 0 \tag{14}$$

which is a second order polynomial that always has a positive solution for $q_b^{t+\Delta t}$.

Figure 1 shows the evolution of $q_v$ and $q_b$ during an idealised sublimation experiment with arbitrarily prescribed initial
conditions. Different numerical resolution methods are tested: *i)* the proposed 'double implicit' method; *ii)* a fully explicit method in which $q_b$ and $q_v$ at the right-hand side of Eq. (11) are treated explicitly; *iii)* a method with an exact resolution of Eq. (11) in $q_b$ - classical linear ordinary differential equation - and explicit treatment of $q_v$ ; and *iv)* an exact resolution in $q_v$ and an explicit treatment of $q_b$. The time step used here is 15 min i.e. the common value used for the LMDZ physics in particular

during CMIP6 (Hourdin et al., 2020). Our 'double implicit' method is numerically stable and it is the closest to the reference curve corresponding to the solution with a 1 s time step.

When blowing snow particles enter an air layer with positive Celsius temperature, we make them melt and evaporate with an exponential decay:

$$\left.\frac{\partial q_b}{\partial t}\right|_{melt} = -\frac{q_b}{\tau_m} \tag{15}$$

using a temperature dependent time scale $\tau_m$ that decreases with increasing air temperature $T$:

$$\tau_m = \tau_{m0} e^{-\frac{T-T_0}{T_m-T_0}} \tag{16}$$

with $\tau_{m0} = 10$ min and $T_m = 278.15$ K. Furthermore, following Gerber et al. (2023) we make all blowing snow sublimate if $q_b < q_{b,min}$ with $q_{b,min} = 10^{-10}$ kg kg$^{-1}$.

Blowing snow particles sediment through the resolution of the sedimentation equation:

$$\left.\frac{\partial q_b}{\partial t}\right|_{sed} = \frac{1}{\rho}\frac{\partial \rho w_b q_b}{\partial z} \tag{17}$$

with $w_b$ the blowing snow settling velocity that we assume constant and equals $w_b = 0.5$ m s$^{-1}$, value that concurs with blowing snow terminal velocity estimations by Mann et al. (2000). It is worth noting that the simulation of the blowing snow flux and net snow erosion is particularly sensitive to this parameter which can be made reasonably varied between 0.2 and 0.6 m s$^{-1}$ depending on the particle size considered. The value of 0.5 m s$^{-1}$ has been set as it gives the most reasonable values of blowing snow fluxes in preliminary simulation tests in Adélie Land (not shown). Eq. (17) is numerically resolved implicitly in time. During their fall, blowing snow particles which initially have the temperature of the overlying layer are 'thermalised' with the ambient air such that the mixture of air and crystals has a unique temperature at each level.

## 2.6 Radiative effects

We take into account the radiative effect of blowing snow through the change of cloud fraction $\alpha_c$ assuming that it scales with the mean-mesh specific content of blowing snow:

$$\alpha_{c,tot} = \min(\alpha_c + \min(\frac{q_b}{q_{bt}}, 1), 1) \tag{18}$$

with $q_{bt}$ the value for which we assume that all the mesh is covered with a blowing snow cloud. This parameter is absolutely not constrained by any observation and it is set arbitrarily to a value corresponding to intense and widespread blowing snow events in our simulations: 1.0 g kg$^{-1}$.

The radiative scheme of LMDZ then considers the total ice water content i.e. the sum of the specific cloud ice water content with the specific blowing snow water content using a common parameterisation of ice crystal effective radius (Madeleine et al., 2020).

# 3 Applications in Antarctica

## 3.1 Model configuration and comparison with in situ observations

### 3.1.1 Simulation configurations

Two ICOLMDZ simulation configurations will be considered in the study. To evaluate the fine-scale performances of ICOLMDZ to simulate the Antarctic katabatic flow and blowing snow, a regional configuration over Adélie Land is first used. The Adélie Land is particularly known for the intense and persistent katabatic winds originating from the interior of the continent (Parish and Bromwich, 2007; Davrinche et al., 2024) and sometimes leading to intense blowing snow events (Amory, 2020; Vignon et al., 2020). This region is also equipped with instrumental systems giving information about blowing snow flux and occurrence and was considered in several studies to evaluate the simulation of blowing snow transport (e.g., Gallée et al., 2013; Amory et al., 2015, 2021; van Wessem et al., 2018). The regional Adélie Land configuration has been set-up in Wiener et al. (2025) and leverages the new limited-area model (LAM) configuration of ICOLMDZ (Raillard et al., 2024). It consists in a domain (Figure 2) with a 20-km horizontal resolution and a 95 $\eta$ vertical level grid of LMDZ with the first model level at $\sim$8 m above the ground in the coastal Antarctic region (Hourdin et al., 2020). The topography is taken from the dataset of Schaffer and Timmermann (2016) which relies on the Bedmap-2 product. The period covered for the LAM simulations is the 2011 year which encompasses the period considered for the evaluation of the blowing snow scheme of the model MAR (January 2011) in Amory et al. (2015). Sea surface temperature, sea-ice cover and lateral forcing are provided by the ERA5 reanalysis (Hersbach et al., 2020).

A second configuration is then used to assess the overall effect of the blowing snow parameterisation once activated in typical climate runs, especially on the Antarctic SMB. It consists in running the global ICOLMDZ model in a so-called 'AMIP' mode meaning that the model is forced with monthly-mean sea surface temperature and sea-ice cover as well as mean aerosols and ozone concentrations. The same 95-vertical grid is employed and we use a horizontal resolution of $\sim$150 km (corresponding to $nbp = 60$ in the Dynamico namelist file). Simulations are carried out over a 5-year period (2000-2004). To ensure a robust comparison between simulations with and without blowing snow and to compare them with contemporary in situ SMB observational data, the wind components are nudged towards the ERA5 reanalysis with a timescale of 6 h. The nudging is applied only in the mid and high troposphere, that is above the hybrid model level corresponding to a reference sea level pressure of 700 hPa not to alter the dynamical interactions between blowing snow and low-level circulation. It is worth mentioning that the additional computational cost of blowing snow mostly comes from the advection of a new water species in the dynamics rather than the treatment of the new parameterizations (surface snow erosion, turbulent transport, sedimentation and sublimation) in the physics part of the model. In the global configuration, this additional cost is about $+4$ %.

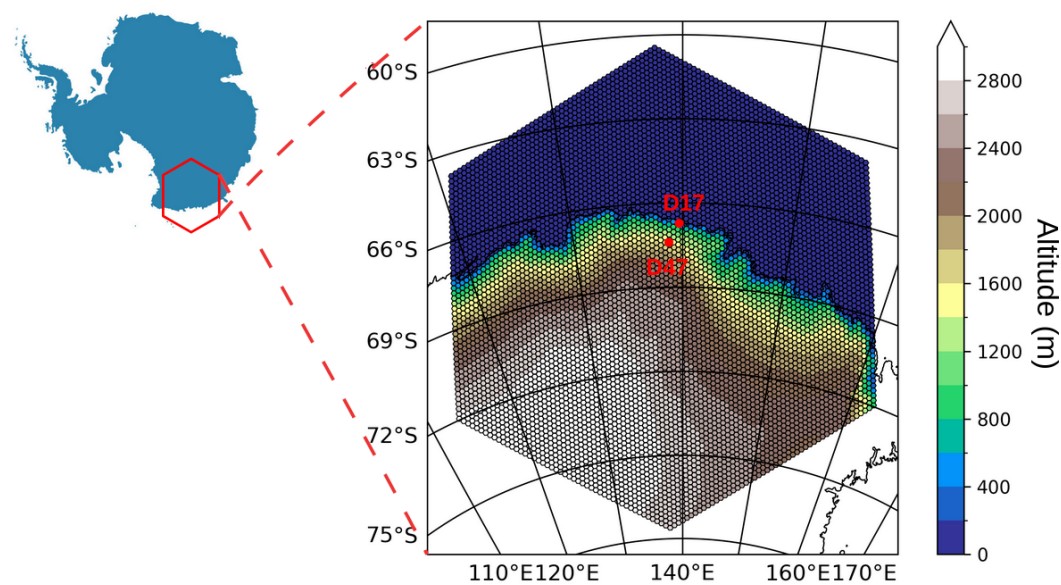

**Figure 2.** Terrain topography in the limited-area simulation configuration grid over Adélie Land. Red dots show the location of the D47 and D17 stations.

### 3.1.2 Observational datasets for model evaluation

In situ measurements of blowing snow are rare due to the remoteness and harsh environment of Antarctica. Active remote sensing retrievals of Antarctic blowing snow from satellite do exist (Palm et al., 2017) and although they provide valuable information at the continental scale, they are quantitatively uncertain and give reliable data in clear-sky conditions, above a height of ≈ 30 m and at a frequency corresponding to the satellite revisit time which make them not always easy to use for quantitative model evaluation. In this study, we leverage a 1-year 2011 time series of in situ measurements collected at the D17 (138.7°E, 67.4°S) and D47 (139.9°E, 66.7°S) stations, located respectively at 10 and 110 km from the coast along a shore-to-Plateau transect between the coastal Dumont d'Urville station in Adélie Land and the inland Concordia station (Figure 2). The topographic channelling of the gravity-driven near-surface flow gives coastal Adélie Land the most intense sustained surface winds on Earth (Parish and Walker, 2006) and very frequent and intense blowing snow events (Amory, 2020; Vignon et al., 2020). This region consists of a sloping snowfield with no major relief but a break in slope at nearly 210 km inland at about 2100 m a.s.l., downstream of which D47 and D17 are located.

At D17, near-surface air temperature, humidity, wind speed are sampled at 6 levels along a 7-m mast (Barral et al., 2014; Amory et al., 2016) while at D47, temperature, humidity and wind are measured at a single level (≈ 2.8 m for wind, ≈ 2.2 m for temperature and humidity) with an automatic weather station (AWS, Amory, 2020). At both stations, meteorological records were complemented with blowing-snow measurements made with 2G-FlowCapt$^{TM}$ sensors. The instrument consists of a 1-m long tube containing electroacoustic transducers that measure the acoustic vibration caused by the impacts of wind-borne snow

particles on the tube. They then provide an estimate of the horizontal snow mass flux – including all forms of wind-driven snow – along the sampling height. In 2011 during our period of interest, two 2G- FlowCapt$^{TM}$ sensors were operating at D47: the first one between 0 and 1 m a.g.l. and the second one between 1 and 2 m a.g.l. At D17, only one 2G- FlowCapt$^{TM}$ installed between 0 and 1 m a.g.l. was operating at this time. The meteorological and blowing snow measurement systems as well as statistics of blowing snow events are extensively presented in Amory (2020). In the present study, we use a processed and formatted dataset described and distributed in Amory et al. (2020). It is worth emphasising that the measurement uncertainty for the 2G-FlowCapt$^{TM}$ is not known. The instrument was shown to generally underestimate the snow mass flux relative to integrated estimates from reference Snow Particle Counters but the sign of the bias reverses when additional precipitation is present. Overall, while the instrument is well suited to detect the occurrence of blowing snow events, the quantification of the blowing snow flux remains quite uncertain and quantitative values should be interpreted with caution. We refer to Amory (2020) (see their Sect. 2.3.3) for an extensive discussion on 2G-FlowCapt$^{TM}$' accuracy and performances. Throughout the year, the lowermost FlowCapt$^{TM}$ gets partially buried due to snow accumulation. At D47, a SR50 acoustic depth sensor monitored the surface elevation continuously between 2010 and 2012 showing that the wind-exposed part of the $H = 1$ m high sensor was $h \approx 0.6$ m in 2011. Building from Amory et al. (2021), the measured flux has therefore been scaled at each time step by $H/h$ to obtain the particle mass flux vertically averaged over the wind-exposed part of the sensor, consistently with the sensor calibration principle which implicitly assumes integration over its full exposed height $H$, requiring correction when only a fraction $h$ is exposed. At D17, the SR50 sensor was deployed in December 2012, thus after the 2011 analysis period considered here. No correction can therefore be applied for this station which likely results in an underestimation of the flux magnitude. As the D17 instruments are raised back manually to original heights at the beginning of each summer field campaign, the underestimation is likely more important during winter and spring but it cannot be properly quantified.

To assess the realism of the Antarctic SMB in global ICOLMDZ simulations, we also use the same SMB observations as in Agosta et al. (2019). Those observations are from the GLACIOCLIM-SAMBA dataset detailed in Favier et al. (2013) and updated by Wang et al. (2016), which follows the quality-control methodology defined by Magand et al. (2007), and from accumulation estimates from Medley et al. (2014), retrieved over the Amundsen Sea coast (Marie Byrd Land) with an airborne-radar method combined with ice-core glaciochemical analysis. We discard observations covering less than 3 years and only keep observations during the 5-year simulation period. We then perform a weighted average – by weighting with the observed accumulation duration – of SMB observations that fall into the same ICOLMDZ grid cell, as in Agosta et al. (2019). At the end, we obtain 308 grid-average accumulation observations.

### 3.1.3   Comparison between observational data and model fields in Adélie Land

Wind speed values $U$ are evaluated at the measurement height $h$ using a common logarithmic extrapolation from the values at the first model level at $z_1 \approx 8$ m and the prescribed roughness length value in the model $z_0$:

$$U(h) = U(z_1)\frac{\log{(h/z_0)}}{\log{(z_1/z_0)}} \tag{19}$$

A similar approach is considered for temperature. At D17, we consider the highest measurement level at $\sim 7$ m a.g.l, i.e. the closest to the first model level height, to limit the influence of the extrapolation. Given the failure of the Monin-Obukhov similarity theory in presence of blowing snow (e.g., Sigmund et al., 2022), the common Monin-Obukhov based humidity interpolation assuming a pseudo-logarithmic profile from surface and first model level values is not adapted. Therefore, relative humidity fields are not vertically extrapolated and direct comparison between first level model fields and observations are shown for qualitative assessment.

The representation of the blowing snow transport will be evaluated through comparison of occurrence and amplitude of the horizontal blowing snow flux defined as:

$$F_b = \rho q_b U \tag{20}$$

with $U$ the horizontal wind speed, $\rho$ the air density and $q_b$ the specific blowing snow content. Note that the 2G-FlowCapts$^{\text{TM}}$ see all type of particles, including snowflakes falling from clouds. However, the LMDZ cloud scheme diagnoses the vertical snowfall flux at each time step but does not compute the specific content – or mass mixing ratio – of snow particles (Madeleine et al., 2020). This prevents us from robustly estimating a horizontal flux of all the particle categories – including snowflakes – from model outputs. While the 2G-FlowCapt$^{\text{TM}}$ provide a mean value over a 1-m height either between approximately 0 and 1 m a.g.l. or between 1 and 2 m a.g.l., the near-surface horizontal flux calculated by the model is by essence a mean value over the full first model layer, which is much deeper than 1 or 2 m.

A direct quantitative comparison of flux magnitude between observations and simulation output is therefore very delicate. One possibility for the D47 site is to compute a mean value over the first model layer depth after a vertical extrapolation of the flux from the measurements of the two superimposed 2G-FlowCapt$^{\text{TM}}$. The vertical profile of the particle mass flux follows an exponential decay in the saltation layer (Martin and Kok, 2017; Melo et al., 2024) which results in an overall exponential decay of the flux with increasing height (Mann et al., 2000; Gordon et al., 2009; Sigmund et al., 2025). In the suspension layer however, the blowing snow concentration and the blowing snow mass flux are expected to be close to a power-law profile of height (Nishimura and Nemoto, 2005). As the lower FlowCapt sensor averages over both the saltation layer and a part of the suspension layer, it is difficult to predict which profile function would be most suitable for extrapolation of the FlowCapt measurements. Although uncertain, an exponential extrapolation of the form $F_b(z) = F_{b0}e^{-z/H_b}$ is used here as a first approach, $F_{b0}$ and $H_b$ being determined with the two 2G-FlowCapt$^{\text{TM}}$ measurements. Note that cases for which the flux at the highest 2G-FlowCapt$^{\text{TM}}$ is stronger than that at the lowermost one have been filtered out. Those cases generally correspond to strong flux values and for which the two measurements are close, and the extrapolation leads to unrealistically large flux values over the first model layer depth. At D17, the presence of one single 2G-FlowCapt$^{\text{TM}}$ in 2011 makes it impossible to apply this method. Nonetheless, the extrapolation method is also very uncertain as it accumulates the measurement uncertainties associated with the two 2G-FlowCapt$^{\text{TM}}$. Be that as it may, quantitative flux magnitude comparison should thus be interpreted with a lot of caution and for D47, both extrapolated and local flux measurements of both sensors in the lowest 2 m will be shown when evaluating the model.

Blowing snow occurrence is evaluated by counting the number of significant blowing snow transport events - at the hourly time step - in both the models and observations. Amory et al. (2021) consider a significant blowing snow event if the hourly-mean flux exceeds a threshold of $1 \text{ g m}^{-2} \text{ s}^{-1}$ . As this threshold was used for fluxes at a 1 m height, we applied the above-explained extrapolation method at D47 to provide an equivalent value for a mean flux integrated over the full first model layer. At D47 a $1 \text{ g m}^{-2} \text{ s}^{-1}$ flux measured by the 2G-FlowCapt$^{\text{TM}}$ between approximately 0 and 1 m value corresponds to a mean of $0.140 \text{ g m}^{-2} \text{ s}^{-1}$ once integrated over the first model layer depth. In the model, we thus assume that there is significant blowing snow event when the hourly-mean intensity of the flux at the first layer exceeds $0.140 \text{ g m}^{-2} \text{ s}^{-1}$. In the observations, we detect a blowing snow event using the 2G-FlowCapt$^{\text{TM}}$ between 0 and 1 m and consider the $1 \text{ g m}^{-2} \text{ s}^{-1}$ threshold.

It is worth emphasising that the comparison between model and observations would be much easier if ICOLMDZ were run with another vertical grid including a first model level at 1 or 2 m a.g.l.. However, we want here to develop and evaluate a blowing snow parameterisation using the standard global climate configuration of the model, for which a very shallow first model layer should be avoided for numerical cost issues. Moreover, changing the vertical grid of the model would require a full re-calibration of the parameterisations - in particular the turbulent diffusion scheme - as a given version of the model 'physics' is a coherent combination of a suite of parameterisations, a vertical grid and a calibration of tuning parameters. In the present study, we deliberately want to evaluate the current version of the model physics operating in ICOLMDZ with its standard physical package and vertical grid.

## 3.2 Evaluation of the parameterisation in Adélie Land

### 3.2.1 Focused analysis on January 2011

The parameterisation is now evaluated using limited area simulations over Adélie Land run over the 2011 year. We start the analysis with a focus on January 2011, the month that served as a test case period for the evaluation of the blowing snow parameterisation in MAR in Amory et al. (2015). Simulation with (respectively without) blowing snow will be refered to as 'BloS' (respectively 'NoBloS'). Figure 3 shows the time series of wind speed, temperature, relative humidity and blowing snow flux at D17 and D47 stations during this period. The overall wind speed evolution is captured by the model at the two stations but a systematic moderate underestimation of strong wind events is noticeable at D47, a bias shared by other models and reanalysis products (Amory et al., 2021; Gerber et al., 2023) and whose origin has not been elucidated yet but may come from a combination of the representation of surface drag (Wiener et al., 2025) and large scale synoptic forcing (Caton Harrison et al., 2024). Temperature evolution is reasonably well reproduced at both stations except a cold bias when the diurnal cycle is particularly well pronounced during the first half of the month. The activation of the blowing snow parameterisation has overall a little effect upon simulated wind and temperature time series. In fact, the moderate blowing snow fluxes and concentrations in January are not sufficiently strong to significantly affect the air temperature and atmospheric stability - and subsequent katabatic forcing - through particles sublimation. Figures 3g and h show that the BloS simulation captures quite well the timing of blowing snow events at both stations. Again, the quantitative comparison of flux magnitude between near-surface

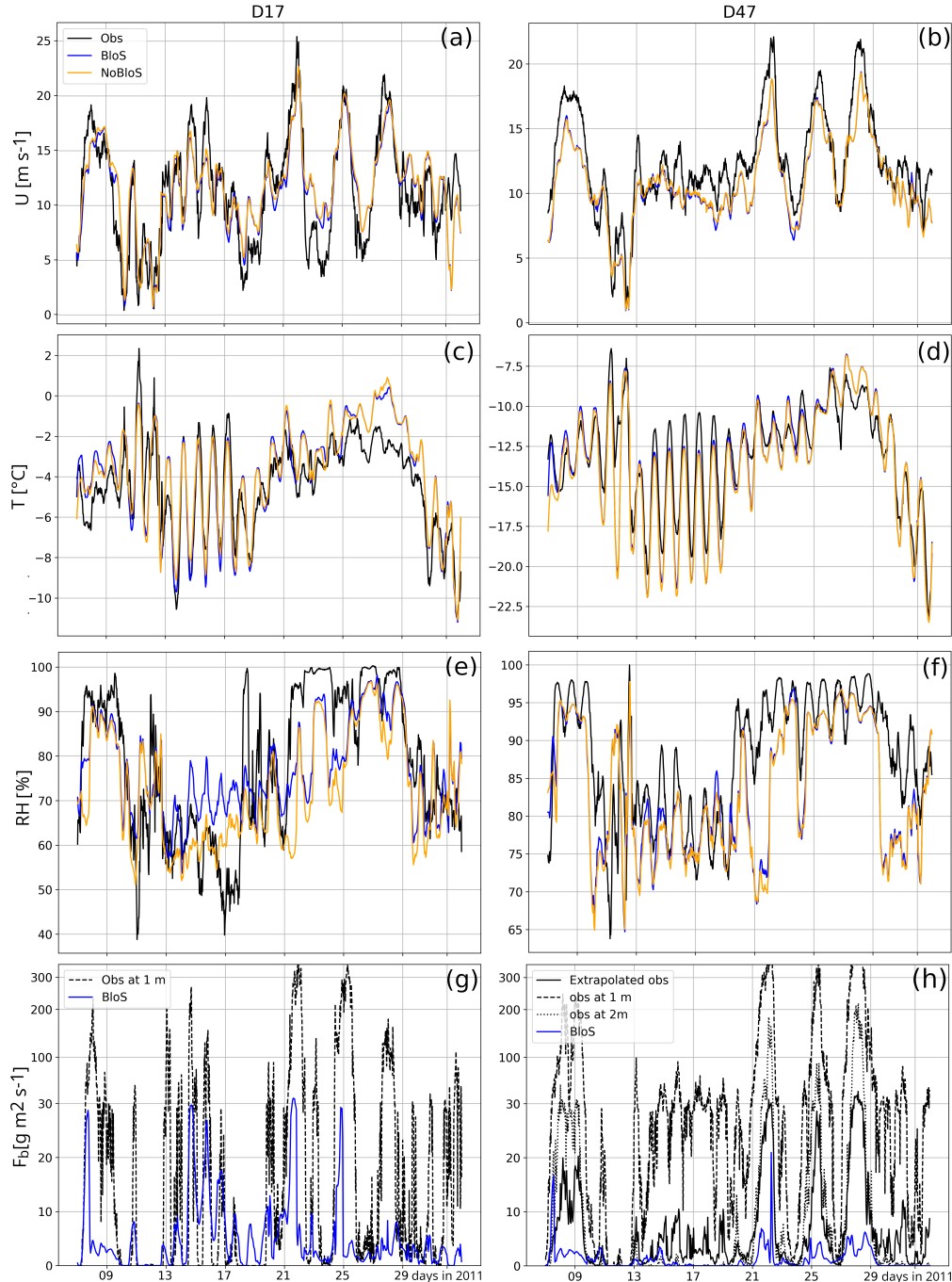

**Figure 3.** January 2011 time series of wind speed (a,b), temperature (c,d), relative humidity with respect to ice (e,f) and blowing snow flux (g,h) at D17 and D47 station. Black lines show in situ observations, orange lines the simulation with no blowing snow (NoBlo) and blue lines the simulation with blowing snow (Blo). At D17, the observed blowing snow flux is that directly measured by the FlowCapt[TM] between approximately 0 and 1 m. At D47, measurements between approximately 0 and 1 m after correcting for the partial burial of the sensor (dashed line), between 1 and 2 m (dotted line) and averaged over the first model layer depth after extrapolation (solid line) are shown. Note the non-linear y-axis in panels g and h.

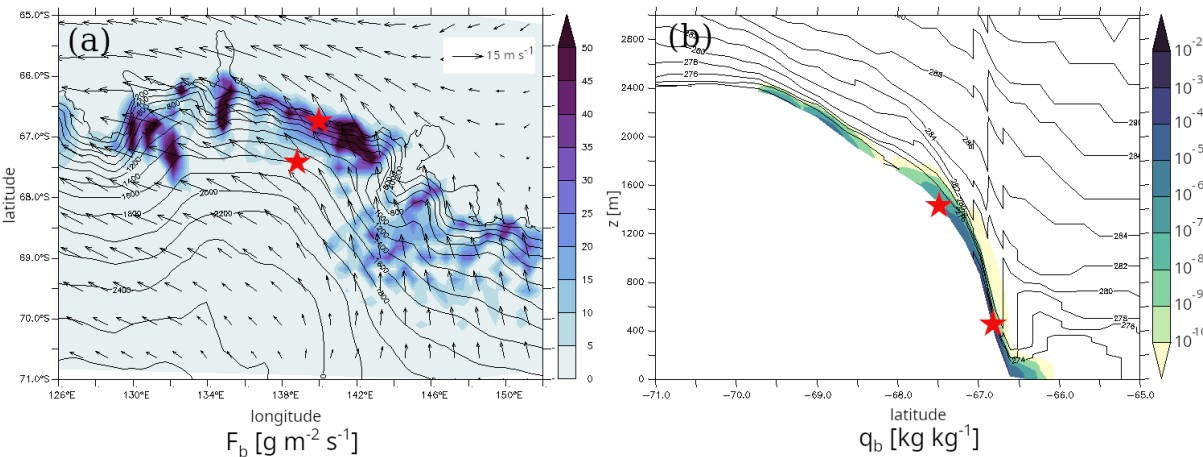

**Figure 4.** (a): map of the blowing snow flux at the first model level at 12:00, 21 January 2011 in the regional BloS simulation at 20-km horizontal resolution. D17 and D47 stations are indicated with red stars. 10-m wind is also plotted with arrows and terrain elevation is shown in black contours (one contour every 200 m). (b): Cross section of the blowing snow concentration at 12:00, 21 January 2011 and at the longitude of D17 (139.9°E). Potential temperature is shown in black contours (one contour every 2 K). Red stars indicate the latitude of D17 and D47 stations.

observations and model output representative of the first model layer is very delicate but the order of magnitudes of the simulated flux is reasonable at the two stations. An underestimation of the simulated flux at D47 compared to extrapolated observations during the 4 main peaks coincides with the underestimation of the wind speed, and is therefore not necessarily attributable to the blowing snow scheme only.

Figure 4a further shows that the modeled blowing snow flux can exhibit a patchy pattern with quite strong spatial heterogeneities, making the local evaluation to station data even more delicate. Such spatial heterogeneities depend on the local wind magnitude but also on the snow density spatial distribution, which itself inherits from the spatial distribution of past snowfall and snow erosion. Figure 4b shows that during wind peaks such as at 12:00, 21 January, the blowing snow layer slightly deepens at the bottom of the slope, where the katabatic jump forms and manifests as near-vertical isentropes. The value of the blowing snow flux in this region which includes D17 thus also depends on the ability of ICOLMDZ to simulate the turbulent mixing associated with the large eddies within katabatic jumps, an aspect discussed in Wiener et al. (2025) and that deserves further work.

Regarding the humidification effect associated with the blowing snow sublimation, Figures 3e and 3f show a moderate effect but the model fails to capture periods of saturation (RH= 100 %) at D17. At D47, part of the overall low RH bias can be explained by the difference between the height of the first model level and that of the measurement ($\approx 2.2$ m). We will see in the next section that a more pronounced humidification signal emerges when considering the full year and especially when including the winter season.

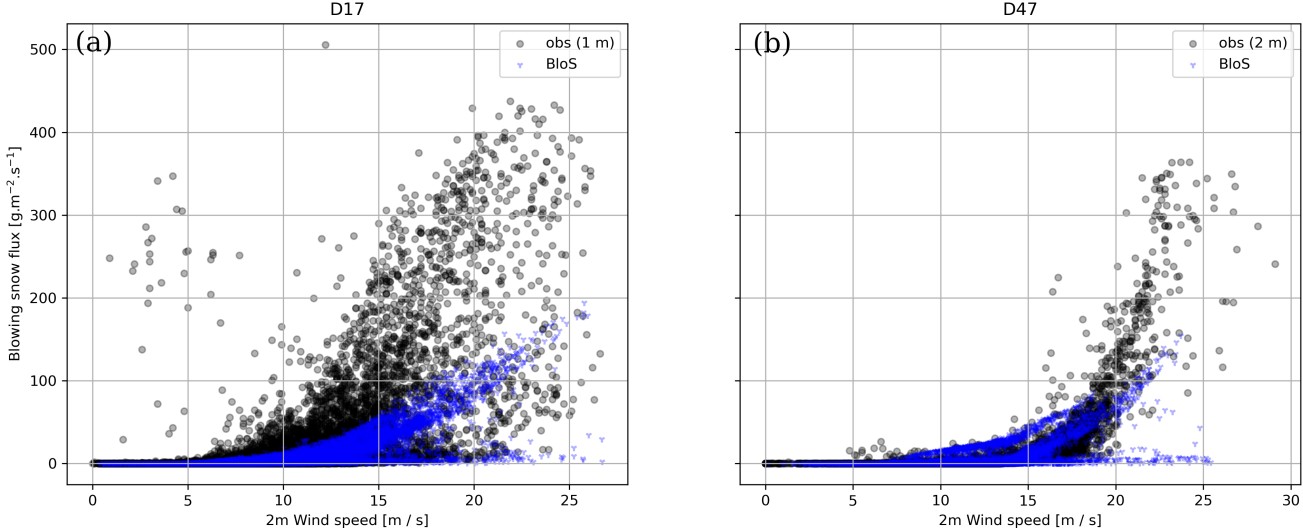

**Figure 5.** Surface blowing snow flux as a function of 2-m wind speed at D17 station (a) and D47 station (b) during the whole 2011 year. Black dots show the 2G-FlowCapt$^{\text{TM}}$ observations (highest sensor at D47) and blue dots show the ICOLMDZ LAM simulation output at the first model level.

### 3.2.2 Yearly statistics in 2011

We now study the full 2011 year and in particular winter months, including stronger wind events and stronger snow erosion events. Figure 5 shows that the relationship between simulated surface wind and blowing snow flux (blue dots) exhibits a hockey-stick behaviour, a pattern that emerges in the observations (see also Amory, 2020) but that can be quite challenging to simulate (see for instance Gadde and van de Berg, 2024). At high wind speed values at D47 and at all wind speed values at D17, the blowing snow flux observations – corresponding either to the single 2G-FlowCapt$^{\text{TM}}$ at D17 and the highest one at D47 – exhibits a more pronounced slope with increasing wind intensity. This might be due to a too strong snow-densification negative feedback in the model or to an overly efficient blowing snow sedimentation. The underestimated increase of the mass flux with wind speed might also be explained by the overly simple saltation model of Pomeroy (1989) considered here, which can affect the predicted relationship between the blowing snow concentration at the top of the saltation layer and the friction velocity. Nonetheless, the direct quantitative comparison between model outputs and observations and the hypotheses that can emerge from it should be interpreted with caution as the simulated flux is representative of the full first model layer. At low wind speed which generally corresponds to situations far from snowfall events or corresponding to weak snowfall events, the model tends to overestimate the flux at D47 which might be attributed to a too slow surface snow densification or excessive simulated snowfall by the LMDZ precipitation scheme, leading to an excess in surface fresh snow.

At D47 (Figure 6b), albeit slightly underestimated, the monthly-mean wind speed is simulated quite reasonably all year long (2011 mean bias=-1.1 m s$^{-1}$, RMSE=3.2 m s$^{-1}$). Monthly mean temperature evolution (Figure 6d), is also well captured

| Station | Model | POD (%) | FAR (%) |
|---------|-------|---------|---------|
| D17 | ICOLMDZ | 94.2 | 37.1 |
|     | MAR | 80.9 | 25.4 |
| D47 | ICOLMDZ | 55.8 | 9.8 |
|     | MAR | 64.5 | 13.4 |

**Table 1.** Probability of detection (POD) and False Alarm Ratio (FAR) calculated at the hourly time scale at D17 and D47 for the 2011 ICOLMDZ LAM simulation and from MAR simulations (numbers from Amory et al. (2021)). Note that the period used in the MAR simulation is longer: January 2010 – December 2012 at D47 and February 2010 – December 2018 at D17.

except that a cold bias is noticeable in the core of the winter. The lack of measurement of surface energy budget components – especially radiative fluxes – at the station prevents us to properly determine the causes of this bias in the model but a possible lack of downward longwave radiative flux in relation with shortcomings in cloud cover and properties might be suspected

(Le Toumelin et al., 2021).

At D17 (Figure 6c) the monthly mean temperature is well captured throughout the year but the most prominent feature is an overestimation of the monthly mean wind speed (2011 mean bias=1.3 m s$^{-1}$, RMSE=5.2 m s$^{-1}$) particularly during winter months (Figure 6a). Such a positive wind speed bias at D17 can also be present in other regional climate models when run at horizontal resolutions greater than $\approx$ 10 km (e.g., Davrinche et al., 2024). Such a bias can be explained, at least partly, by

435 an underestimation of the magnitude of the so-called 'shallow baroclinicity' or 'thermal wind' forcing that acts to slow down the low-level outflow at the coast (Caton Harrison et al., 2024; Davrinche et al., 2024). This is particularly pronounced at low horizontal resolution for which horizontal gradients of potential temperature are smoother, resulting in a overly smooth and too far downstream 'katabatic jump' (Vignon et al., 2019).

The magnitude of the simulated blowing snow flux at the first model level at D17 is either close to or even exceeds the 2G-

440 FlowCapt$^{\text{TM}}$ measurements between 0 and 1 m (Figure 6e) and is therefore likely overestimated, concurring with the too strong simulated wind speeds at this station, particularly during the extended winter. At D47, the blowing snow flux intensity exhibits more reasonable values compared to observations (Figure 6f) even though the July value - very close to the FlowCapt$^{\text{TM}}$ measurements between 0 and 1 m - is likely overly strong.

In terms of blowing snow occurrences, Figure 7 shows an overestimation throughout the year at D17 - which coincides with

445 the overestimation in wind speed - while the simulated frequency is more realistic in July, August and December at D47 but is underestimated the rest of the year. The near-persistent blowing snow in winter in the simulation leads to a high probability of detection (POD) but also to a quite high false alarm ratio (FAR) at D17 (Table 1). The POD is lower at D47 (58.5) but the POD/FAR ratio at D47 is comparable with that reported for the MAR model in Amory et al. (2021) albeit over a different period.

Including the blowing snow parameterisation modifies the overall year-averaged structure of the boundary layer in coastal Adélie Land. Figure 8b depicts the moderate near-surface cooling of the boundary layer especially at the bottom of the slope, mainly explained by the latent heat effect associated with blowing snow crystals sublimation (Hofer et al., 2021). The latter also

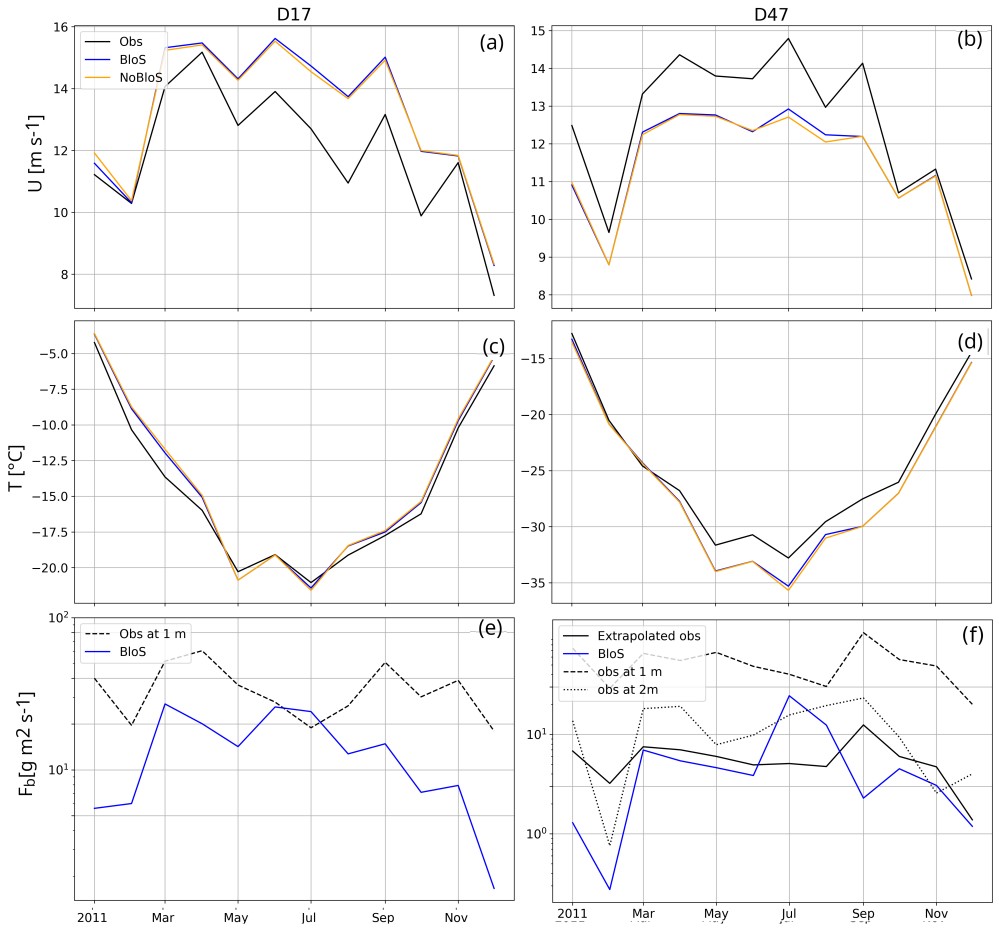

**Figure 6.** 2011 time series of monthly mean wind speed (a,b), temperature (c,d) and blowing snow flux (e,f) at D17 and D47 station. Black lines show in situ observations, orange lines the simulation with no blowing snow (NoBloS) and blue lines the simulation with blowing snow (BloS). At D17, the observed blowing snow flux is that directly measured by the FlowCapt[TM] between 0 and 1 m. At D47, measurements between 0 and 1 m (dashed line), between 1 and 2 m (dotted line) and averaged over the first model layer depth after extrapolation (solid line) are shown. Note the differences in y-axis range between the left and the right columns as well as the non-linear y-axis in panels e and f.

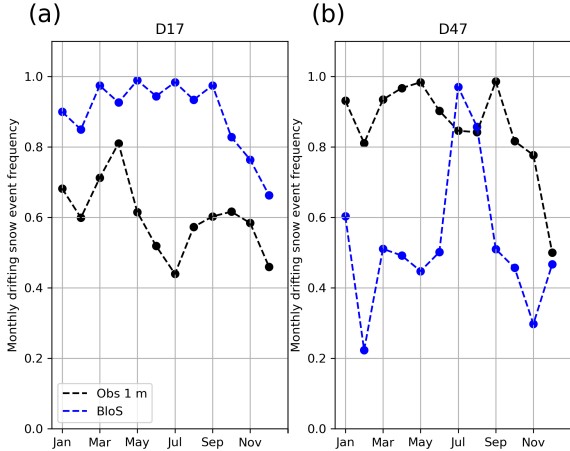

**Figure 7.** Monthly frequency of blowing snow occurrences in observations (black) and ICOLMDZ LAM simulation outputs (blue) at D17 (a) and D47 (b).

leads to a pronounced humidification - exceeding 10 % in relative humidity locally - in a thin layer to the ground surface and extending even offshore (Figure 8d). The investigation of differences in surface energy budget contributions reveals an overall
increase in the yearly averaged downward longwave radiative flux at the surface (LWdn) due to the presence of the blowing snow cloud (Figure 8e). However, this increase in LWdn is partly compensated by a weak decrease in the yearly averaged net shortwave flux (SWnet) due to the sunlight reflection by blowing snow crystal as well as by a decrease in surface turbulent sensible heat flux ($H_s$) due to the cooling of near-surface air. Such findings are in agreement with a similar investigation using the MAR model in Hofer et al. (2021). The increase in near-surface relative humidity when the blowing snow scheme
is activated leads to a weak decrease in the magnitude of the surface turbulent latent heat flux $H_l$ - which only accounts for the sublimation of surface snow - at the continental margins (Figure 8e). Overall, the inclusion of blowing snow leads to a limited net surface warming (Figure 8f). For example at D47, the increase in LWdn reaches $+8.3$ W m$^{-2}$ while the decrease in SWnet and $H_s$ equal $-1.2$ W m$^{-2}$ and $-6.5$ W m$^{-2}$ respectively, leading to an overall increase in yearly averaged surface temperature of 0.2 K.

## 3.3 Antarctic SMB impact in global simulations

The effect of the blowing snow parameterisation is now assessed in global runs using the horizontal and vertical resolution of the model chosen for the upcoming CMIP7 exercise. No major change is observed at the global scale in terms of temperature and humidity fields outside of the two main ice sheets (not shown). On the Antarctic ice sheet, an increase by several percent in cloud cover is observed on the periphery which is mostly explained by the presence of blowing snow clouds (Figure 9a). In
accordance with the results obtained in Adélie Land simulations, an overall increase in near surface relative humidity is also observed along the periphery (Figure 9b) due to blowing snow sublimation, concurring with the results of Gadde and van de

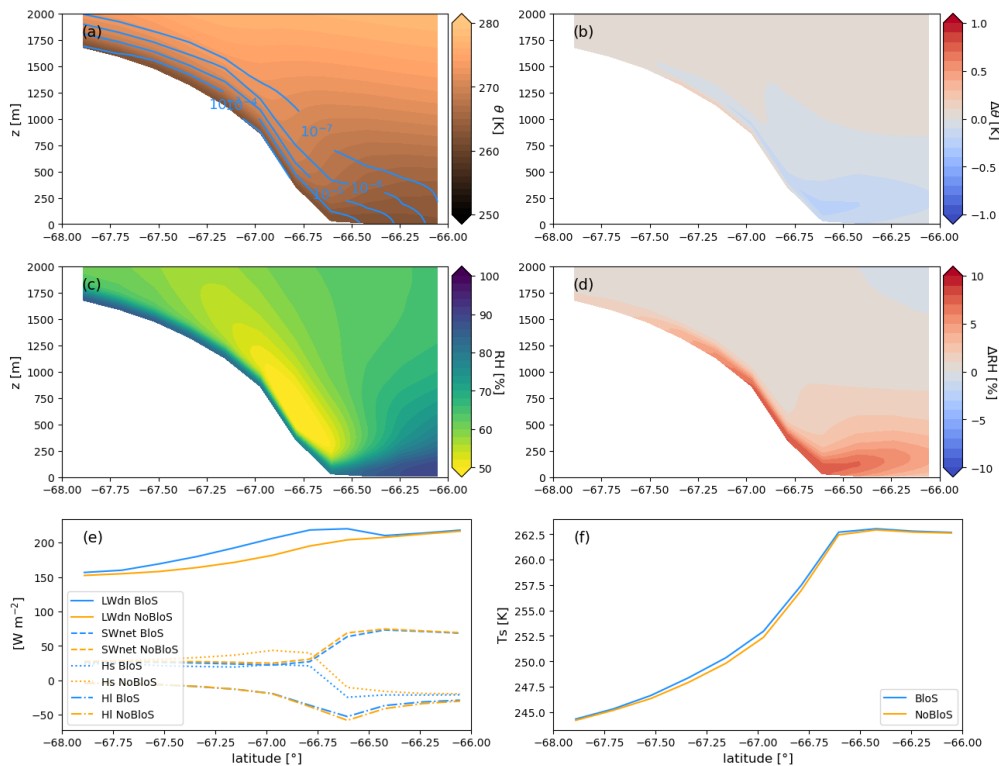

**Figure 8.** Yearly averaged zonal cross-sections at 140.0° from the LAM simulations (a) potential temperature (in K, shading) and $q_b$ (in kg kg$^{-1}$, contours) in the BloS simulation. (b): difference in potential temperature between the BloS and NoBloS simulations. (c): Relative humidity wrt ice in the BloS simulation. (d): Difference in relative humidity between the BloS and NoBloS simulations. (e): Downward longwave radiative flux (solid lines), Net shortwave surface radiative flux (dashed line), surface turbulent sensible heat flux (dotted line) and surface turbulent latent heat flux (dash-dotted line) in the BloS (blue) and NoBloS (orange) simulations. Fluxes are defined positive toward the surface. (f) surface temperature in the BloS (blue) and NoBloS (orange) simulations.

Berg (2024) (see their Figure 7c). A slight warming of surface temperature and cooling of 2 m temperature reaching a few tenths of K is also noticeable along the Antarctic periphery when including blowing snow (not shown), again in agreement with the results obtained previously over Adélie Land. We now conduct an analysis on the impact of our blowing snow

parameterisation on the Antarctic SMB in global runs. As the Greenland SMB is very affected by surface snow melting and refreezing processes for which the parameterisations in LMDZ is still very crude, we leave the analysis of the Greenland SMB for further research leveraging the coupling with the ORCHIDEE land surface model including an advanced representation of surface snow processes over ice sheets (Charbit et al., 2024).

Figure 10a shows the simulated Antarctic SMB averaged over the 5 years of simulations. Comparison with observations

(circles) reveals a reasonable agreement, except to the east of the Peninsula. This might be attributed to an excess of precipitation associated with a possible underestimated Foehn effect due to the quite coarse horizontal resolution employed in the global runs.

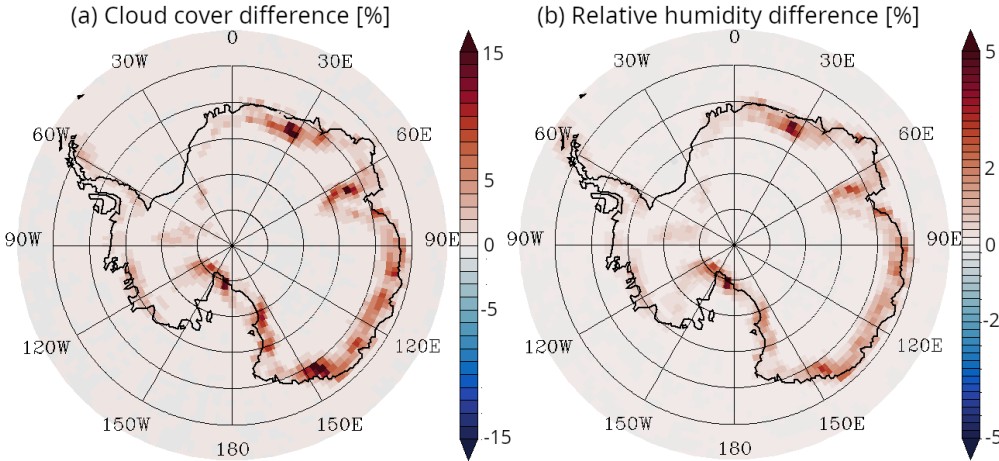

**Figure 9.** Difference in total cloud cover (a) and relative humidity at the first model level (b) between the global BloS and NoBloS simulations. Averages over the full simulation length are shown.

Figure 10b shows that accounting for blowing snow overall increases the SMB along the East-Antarctic coast and decreases its value in the escarpment region, a few tens to hundreds km inland. The difference can locally reach several tens of $\mathrm{kg\ m^{-2}\ yr^{-1}}$ but the absence of SMB measurements in the regions with the strongest changes prevents us from concluding about a possible improvement or deterioration of the local SMB modelling. Two hypotheses can be proposed for the coastal increase in SMB: i) an increase in blowing snow deposition associated with the surface snow erosion upstream and ii) an increase in snowfall flux associated with the humidification of the boundary layer by blowing snow sublimation (Lenaerts and van den Broeke, 2012) that weakens the precipitation sublimation effect in the katabatic layer (Grazioli et al., 2017; Jullien et al., 2020). Figure 11 shows that the two effects are at play. Along the 90°E-135°E sector, the black line shows the effect of the erosion-deposition process due to the blowing snow parameterisation leading to a decrease in SMB near $\approx 68°$S latitude and an increase closer to the coast. The red line further reveals an increase in snowfall (that does not include the sedimentation flux of blowing snow) between the BloS and NoBloS simulation. Careful inspection of the vertical profiles of snowfall reveals similar snowfall values in altitude in the two simulations - suggesting no overall increase in large-scale precipitation amount in altitude - but differences close to the surface where sublimation in the katabatic layer occurs (not shown).

## 4  Discussion and conclusions

Recent regional model findings suggest that the aeolian erosion of surface snow is a significant contribution to the overall Antarctic SMB through ice crystals sublimation and export outside of the ice sheet. Such findings raise the question of the relevance of accounting for such a process even in global climate models. This paper presents the development and evaluation of an intermediate-complexity blowing snow parameterisation for the ICOLMDZ AGCM, atmospheric component of the

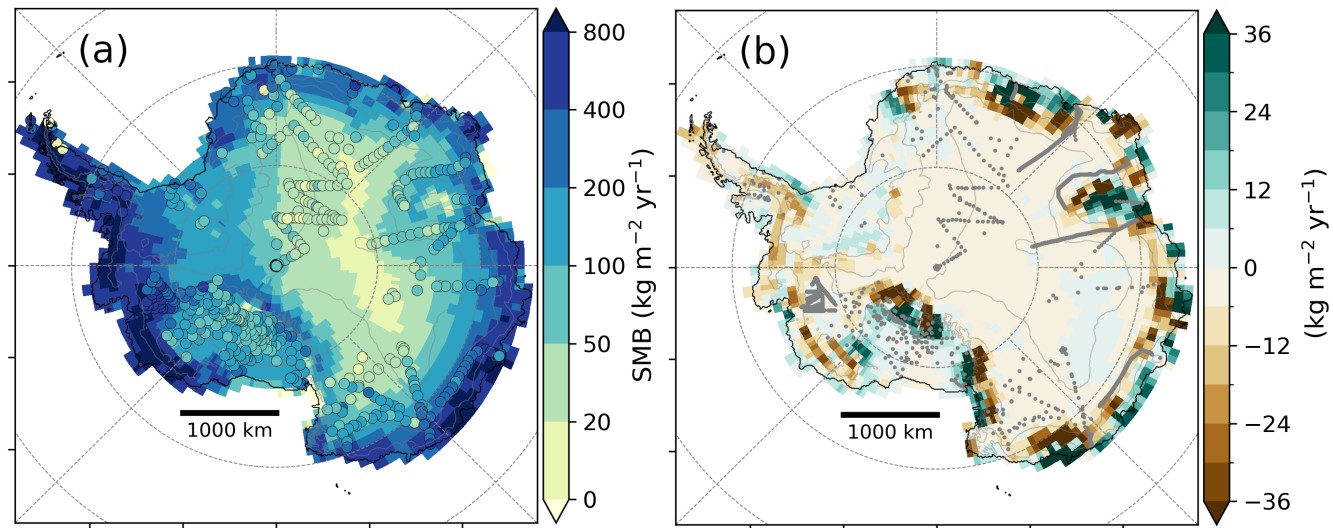

**Figure 10.** Simulated Antarctic SMB in the global simulations: (a) 2000-2004 mean annual SMB in the global simulation with blowing snow, with coloured dots showing the observed SMB values (shared colour scale). Simulated SMB is plotted only for pixels for which the land-ice fraction exceeds 30%. (b): difference in mean annual SMB between the global simulation with and that without blowing snow. Grey dots in panel (b) show the location of all SMB observations available in the observation dataset. Circles in panel a are the averaged values from observations within each model grid cells, the average being calculated by weighting with the observed accumulation duration. Let's recall that we discard observations covering less than 3 years and only keep observations during the 5-year simulation period.

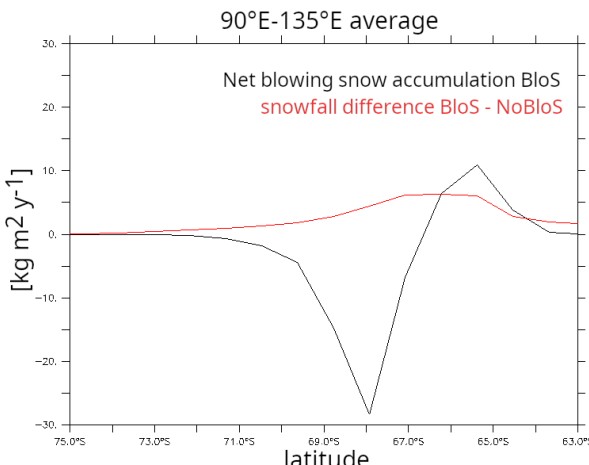

**Figure 11.** Black line: zonal evolution of the annual mean net blowing snow accumulation (deposition - erosion) in the global simulation with blowing snow (Blos). Red line: zonal evolution of the annual mean snowfall difference between global simulations with (BloS) and without blowing snow (NoBloS). We consider here the geographical average over an East-Antarctic sector between 90°E and 135°E and the temporal average over the 2000-2004 period.

IPSL Coupled Model. The parameterisation is inspired by that implemented in the MAR model, but the specific content of blowing snow is treated as an independent water species. We try to find a reasonable trade-off between parameterisation sophistication and applicability in the AGCM, implying particular attention to the numerical stability and numerical cost. The behaviour, performance and effect of the parameterisation are first assessed in limited-area simulations over Adélie Land. We deliberately keep the standard physical package and vertical resolution used in global climate simulations, although the quantitative comparison with in situ measurements becomes even more delicate. In January, when the model captures fairly well the temperature and wind speed along the Adélie transect, simulated snow flux occurrences are very well captured . The order of magnitude of the flux is also fairly well reproduced but the moistening effect of the surface layer is underestimated during moderate transport events likely due to underestimated blowing snow sublimation. During winter at D47, the monthly mean wind speed is overestimated by about $1 \mathrm{~m~s}^{-1}$ and a mean cold bias ranging between 1 and 2 K is noticeable. The snow flux occurrence fits well the observations in July and August but the amplitude is probably overestimated. The hockey-stick relationship between blowing snow flux and wind speed is captured by the model, and the probability of detection and false alarm ratio are similar when compared to MAR performances. Closer to the coast at D17, the simulated wind speed is overestimated, a bias also present in other regional climate models and that questions the representation of the location and intensity of the 'katabatic jump' when surface wind speed over the ice sheet is strong. Such overly strong winter winds coincide with an overestimation of wintertime occurrences of blowing snow near the coast. The effect of the blowing snow scheme on the mean temperature and relative humidity fields is also quantified, with the most prominent feature being the moistening of the first tens of meters above the ground due to blowing snow sublimation. The impact of blowing snow on the simulated surface energy budget is also analysed, but the overall effect on the mean surface temperature is quite weak due to a compensation between the increase in downward longwave radiative flux and the decrease in net shortwave radiative flux and turbulent sensible heat flux.

The effect of the blowing snow scheme is then assessed in global climate simulations with a particular focus on the Antarctic climate and SMB. With respect to the simulation with no blowing snow, an increase in cloud cover and near surface relative humidity is noticeable along the Antarctic periphery and a significant increase (reps. decrease) in SMB is simulated along the East Antarctic coast (resp. escarpment region). The latter is explained by both erosion deposition process along the near-surface outflow and by the increase in snowfall due to a weakening of the low-level snowflake sublimation in response of the moistening effect associated with blowing snow sublimation. The difference is locally not negligible as it can exceed several tens of $\mathrm{kg~m}^{-2} \mathrm{~yr}^{-1}$ in magnitude.

The overall significant impact of blowing snow on simulated SMB and coastal surface energy exchanges, combined with its very limited influence on the climate at lower latitudes, are strong arguments in favor of including blowing snow processes in global climate model simulations, particularly in configurations focusing on polar regions, such as those coupled with ice sheet models (e.g., Smith et al., 2021) or aiming at a more comprehensive representation of the atmospheric water cycle. However, this statement should be nuanced. First, our study has not demonstrated a systematic improvement in simulated radiative fluxes or SMB compared to observations, highlighting the need for further evaluation of those quantities at the Antarctic scale. Second, including blowing snow adds a modest computational cost ($\approx 4\%$), which may become a limiting

factor for long-term simulations or ensemble experiments, especially when increasing model resolution to better capture the spatial variability of precipitation over the ice sheets. We therefore recommend the use of blowing snow parameterizations in global climate models in experiments specifically targeting polar processes or aiming to better represent the hydrological coupling between the atmosphere and the cryosphere. Nonetheless, further evaluation is required to confirm that the additional process representation in ICOLMDZ leads to improved model performance at the Antarctic scale.

While this first version of blowing snow parameterisation in ICOLMDZ is conclusive to some extent, several research questions and avenues for improvement can be raised. It is first worth noting that the parameterisation contains a number of parameters that remain to be more robustly constrained, such as the value of specific content of blowing snow $q_{bt}$ which determines the fraction of the mesh covered with blowing snow for radiative transfer calculations, or parameters that determine the feedbacks of blowing snow, rainfall and melt onto snow density. Advanced model tuning methodologies could be leveraged (e.g., Hourdin et al., 2021) but they require reliable and extensive observational datasets of snow properties over well constrained and reference case studies in presence of blowing snow. Then, we expressed the concentration of particles in the saltation layer $q_{b,salt}$ using a formula from the saltation model of Pomeroy (1989) in which the particle mass flux in the saltation layer is assumed uniform in height. Such a model is in contradiction with the well-documented exponential decay of the particle mass flux. Other common saltation layer parameterisations can be used (e.g., Sharma et al., 2023) but they also suffer from physical inconsistencies which make the prediction of the concentration in the saltation layer an active field of research (Melo et al., 2024).

Implementing a blowing-snow parameterisation in an AGCM where the first model layer typically lies several meters above the surface, limits the ability to resolve strong vertical gradients of blowing snow properties near the ground. Refining the vertical discretisation to better capture these gradients would, however, entail a substantial increase in computational cost. Consequently, the formulation of the surface drag coefficient becomes particularly critical in such models, and accounting for subgrid-scale vertical variability in blowing-snow mass content and wind speed may be necessary to improve the representation of blowing snow transport. The parameterisation of the drag coefficient for the mass transfer of blowing snow particles between the saltation layer and the first model level as well as that for the turbulent diffusion in the atmosphere – in our case, the $\zeta_b$ parameter – has been little studied and probably underappreciated hitherto. When aeolian snow particles are present in the surface layer, the standard Monin-Obukhov similarity theory commonly used to compute surface drag coefficients for heat and water vapor as well as to diagnose temperature, humidity and wind in the surface layer is no longer valid. This leads to substantial biases in the prediction of surface heat and vapor fluxes (Sigmund et al., 2022). Further work on the parameterisation of surface fluxes and turbulent diffusion in presence of drifting and blowing snow is therefore needed, along with additional surface fluxes observations during blowing snow events in Antarctica to guide parameterisation developments. We also acknowledge here the genuine added value of meteorological masts with blowing snow measurements in Antarctica such as those presented in Nishimura et al. (2024). Such masts make it possible to compare atmospheric variables almost at the same height as the AGCM first level, regardless of any surface-layer interpolation function. Advanced measurements of blowing-snow (with latest generation FlowCapt$^{\text{TM}}$ FC4 and Snow Particle Counters) are now being collected along meteorological

mast at several sites along the Adélie Land transect in the framework of the AWACA project (https://awaca.ipsl.fr/en/atmospheric-water-cyc
opening avenues for more extensive and accurate blowing-snow parameterisation's evaluation work. Last but not least, some
work is underway to couple LMDZ with the ORCHIDEE land surface model over ice sheet surfaces. The recent version of
ORCHIDEE indeed includes an advanced multi-layer snow parameterisation adapted for ice sheet surfaces (Charbit et al.,
2024) including a snow densification scheme much more elaborated than the heuristic approach proposed here. Future work
should thus complement the ORCHIDEE snow module with a snow erosion scheme as that developed in the present paper.

. *Code availability* The LMDZ and DYNAMICO codes are freely distributed under the CeCILL license. The exact version of the model
used to produce the results is archived in a repository under DOI (https://doi.org/10.5281/zenodo.17531181), along with the input data and
scripts required to run the model and generate the plots for all the simulations presented in this paper (Vignon, 2025).

. *Data availability* The meteorological and drifting-snow data at D17 and D47 can be downloaded at https://doi.org/10.5281/zenodo.3630497
(Amory et al., 2020).

. *Author contributions* EV: parameterisation development, method, supervision, writting (original draft). NC: evaluation, method, analysis,
visualisation, writting (original draft). CA: conceptualisation, simulation analysis, review and editing. CA: conceptualisation, analysis, review
and editing. VW: method, simulation setup, review and editing. JC: results analysis, review and editing. TD: funding acquisition, review and
editing. CG: funding acquisition, review and editing.

. *Competing interests* The authors declare they have no competing interests

.

. *Acknowledgements* All the members of the informal 'groupe neige' are gratefully acknowledged for fruitful discussions. This project has
received funding from the European Research Council (ERC) under the European Union's Horizon 2020 research and innovation programme
(grant no. 951596) through the AWACA project. We acknowledge support from the DEPHY research group, funded by CNRS/INSU and
Météo-France, as well as from the PEPR TRACCS project (no. ANR-22-EXTR-0008) funded from the Agence Nationale de la Recherche -
France 2030. Yann Meurdesoif is gratefully acknowledged for the support to implement the blowing snow in DYNAMICO as well as Patryk
Kiepas for the development of softwares to prepare forcing files for the model. The simulations were granted access to the HPC resources
of IDRIS and TGCC under the allocations gen15038, WUU AD010111116R2 and LMD AD010107632R3 attributed by GENCI (Grand
Equipement National de Calcul Intensif). Meteorological and blowing snow observations at D17 and D47 are obtained and made available

by the CALVA project (https://web.lmd.jussieu.fr/~cgenthon/SiteCALVA/) with the support of the French polar Institute (IPEV project 1013).

We finally thank three anonymous reviewers whose insightful comments helped improve an earlier version of this manuscript.

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
