# Peer review of "Intermediate-complexity Parameterisation of Blowing Snow in the ICOLMDZ AGCM: development and first applications in Antarctica"

_EGUsphere, 2025_

## Author Comment (AC1)

**Revision of**

**'Intermediate-complexity Parameterisation of Blowing Snow in the ICOLMDZ AGCM: development and first applications in Antarctica'**

Etienne Vignon, Nicolas Chiabrando et al.

October 30, 2025

This document contains the response to a review of 'Intermediate-complexity Parameterisation of Blowing Snow in the ICOLMDZ AGCM: development and first applications in Antarctica' submitted to EGUSPHERE for possible publication in Geoscientific Model Development. Comments the Reviewer are in black and answers are in blue. Paragraphs that have been added or modified during the revision process are copied in purple.

**Reviewer #1**

This study aims at developing and evaluating a parameterization of wind-driven snow transport for global climate simulations using the atmospheric general circulation model ICOLMDZ. The parameterization approach is similar to that in the regional atmospheric model MAR. However, the mass mixing ratio of blowing snow is a separate prognostic variable to distinguish blowing snow from precipitation. Additionally, a double-implicit numerical method is proposed to compute blowing snow sublimation, leading to stable and accurate results despite the large time step needed for global simulations. Using a limitedarea simulation over one year, the blowing snow parameterization is compared with in-situ measurements of blowing snow (FlowCapt sensors) and standard meteorology at two Antarctic sites. The frequency and timing of blowing snow events are similarly well reproduced as in the MAR model although the modeled event frequency differs from the measured one by a factor of two in some months. A direct quantitative comparison of the horizontal mass flux of blowing snow is not possible as the measurements cover the lowest one or two meters of the atmosphere while the first grid level of the model is at a height of approximately 8 m. Nevertheless, the authors extrapolate linearly the measured mass fluxes to estimate the vertically averaged mass flux in the layer corresponding to the first model layer, at least at one site. By comparing this estimate with the model result at the first grid level, they conclude that the modeled particle mass flux has a reasonable order of magnitude. Finally, the authors use a model set-up for global simulations and show that blowing snow clearly decreases (increases) the surface mass balance in the escarpment zone (at the coast) of East Antarctica.

We gratefully thank the Reviewer for the thorough and insightful review of our manuscript. We truly appreciated all the comments, which have substantially helped us improve the study. Please find below our detailed responses to each comment.

**General comments**

This paper addresses relevant questions as blowing snow is a wide-spread and frequent phenomenon in Antarctica and other snow-covered regions. The text is generally well structured. The results are well illustrated by figures and mostly discussed appropriately.

- (1) As emphasized by the authors, however, a quantitative validation of the modeled intensity of blowing snow is very challenging due to the coarse vertical grid resolution. I see the following problems with the model-measurement comparison and consequently with the conclusion that the '[blowing-snow flux] amplitude is also fairly well reproduced' (1.456):
- (a) The horizontal mass flux of blowing snow does not decrease linearly with height. The authors mention the exponential decay in the saltation layer (l. 313). In the suspension layer, the mass flux is also expected to decrease nonlinearly with height as the particle concentration can be approximated by a power-law function (e.g., Gordon et al., 2009; Mann et al., 2000; Sigmund et al., 2025) and the wind speed and particle speed profiles by logarithmic functions. Therefore, the linear extrapolation used in the present study appears inappropriate. Please see our answers to your comment 2a.
- (b) The authors consider the mass flux of blowing snow simulated at the first grid level as 'a mean value over the full first model layer' (l. 309). However, as the mass flux decreases non-linearly with height, the mass flux at the first grid level (center of the grid layer) is expected to be lower than the mean value over the first grid layer. Even if the observation-based mean value over this layer was accurate and the same value was modeled at the first grid level, it would imply an overestimation of the particle mass flux and concentration in the model, which would propagate to higher grid levels through the diffusion-sedimentation equation.

We agree with this rationale. The mass specific content of blowing snow at the first model level is by definition - such as all scalar variables of the model - the mean value within this layer. Then one can raise the following questions: Should we consider a subgrid vertical distribution of the blowing snow content (same thing for wind speed) for calculating i) turbulent diffusion and sedimentation in the physics of the model ii) the transport of blowing snow (through mass fluxes) in the dynamics? Such questions are not trivial at all and they can

be neglected when refining sufficiently the vertical discretization of the model (not only in the physics but also in the dynamics). However properly tacking this issue would be a considerable amount of work, that goes well beyond the scope of this paper and that could be generalized to any scalar variable that is supposed to vary vertically at the subgrid scale (this is for example the subject of some literature on the treatment of turbulent diffusion at the stratocumulustopped boundary-layer inversion). The first point would imply rethinking the full numerical resolution of the turbulent diffusion in the model [14], that considers that scalar variables are defined at the middle of the layers. For the dynamics-related aspect, this would imply re-deriving the transport equation which is a huge theoretical and numerical work that might be generalized to any scalar variable with a subgrid vertical variability (common numerical advection schemes such as those developed by Van Leer assume some horizontal variability that can be described with polynomials). Therefore, we unfortunately cannot properly address your comment as we deem it beyond the scope of the present paper and even beyond the modeling of blowing snow. However, we have modified a paragraph in the conclusions to raise this important issue: Implementing a blowing-snow parameterisation in an AGCM where the first model layer typically lies several meters above the surface, limits the ability to resolve strong vertical gradients of blowing snow properties near the ground. Refining the vertical discretisation to better capture these gradients would, however, entail a substantial increase in computational cost. Consequently, the formulation of the surface drag coefficient becomes particularly critical in such models, and accounting for subgrid-scale vertical variability in blowing-snow mass content and wind speed may be necessary to improve the representation of blowing snow transport.

(c) l. 271 - 273: Did the measurement sites experience net snow accumulation and did the lower FlowCapt sensor get burried gradually during the course of the year?

Please see our answers to your comment 2b.

- (2) To improve the model-measurement comparison and increase the confidence in the blowing snow parameterization, I have the following suggestions:
- (a) Instead of extrapolating the measured mass fluxes, one could estimate model-based vertical profiles of blowing snow concentration and wind speed between the snow surface and the first grid level, using the parameterization assumptions and findings from the literature. For example, the particle concentration can be interpolated between the saltation-suspension interface and the first grid level, using a power-law function of height (e.g., Gordon et al., 2009) or a logarithmic profile as implied by the bulk flux parameterization for the vertical blowing snow flux at the lower boundary (Eq. 5). By multiplying the particle concentration and wind speed profiles, it is possible to estimate the mean particle mass flux over the layer covered by the FlowCapt sensor(s) and achieve a more consistent model-measurement comparison.

Thank you very much for this comment and comment 1a which make us reconsider how we compare observed and modelled blowing snow fluxes and in particular how we assume the blowing snow flux to decrease with increasing height. We now have to assume an exponential decay of the flux even in the suspension layer and there is two ways of proceeding:

- 1. Computing a near-surface profile of the modelled blowing snow flux assuming a logarithmic wind profile and an exponential profile of blowing snow mass concentration (your suggestion);
- 2. Assuming an exponential decay of the flux  $F(z) = F_0^{-z/H}$  and computing the mean flux over the first model layer using this relation with observations ( $F_0$  and H being estimated from the two FlowCapts measurements à 0.5 and 1.5 m). This is equivalent to the current methodology but assuming an exponential rather than a linear decay of the blowing snow flux.

The first method is unfortunately not feasible in an offline way (i.e. from model outputs) because we do not have access to the particles concentration in the saltation layer at each time step. This is because surface snow density - and therefore our estimate of erosion threshold and mass concentration in the saltation layer – can change within one time step when the fresh snow is completely eroded. We therefore adopted the second method. Section 3.1.2 has been modified to account for these changes and figures 3, 5, 6, 7 and Table 1 have been updated. Note that the overall results' interpretation and conclusions are unchanged. The paragraph describing our extrapolation method in Sect. 3.1.3 has been rephrased as follows:

'The vertical profile of the particle mass flux follows an exponential decay in the saltation layer [8, 9] which results in an overall exponential decay of the flux with increasing height [7, 5, 13]. An exponential extrapolation of the form  $F_b(z) = F_{b0}e^{-z/H_b}$  is therefore used,  $F_{b0}$  and  $H_b$  being determined with the two 2G-FlowCaptTM measurements'

(b) If the FlowCapt sensor was partially burried and changes of surface elevation were monitored, it would be best to scale the FlowCapt measurement of the partially-burried sensor to obtain the particle mass flux vertically averaged over the wind-exposed part of the sensor. The model-based estimate can be averaged over the same height range, which changes with time.

Thank you for raising this important point which has also been noticed by the other referees. Indeed the lowermost FlowCapt sensor regularly gets partially buried – as illustrated in Figure 1 – and the accumulated snow height can be estimated thanks to a SR50 depth acoustic sensor. However, the SR50 was deployed in December 2012 at D17, and only few information about surface elevation is available in 2011, that is during the analysis period considered in the present study. In fact, the station the instruments are raised back manually to original heights at the beginning of each summer field campaign so the flux is likely subject to an underestimation especially in winter and spring. Unfortunately, no scaling correction can be properly applied on D17 data. At D47, as the SR50 was operational throughout the 2011 year, we apply the same correction as in

Amory et al. [3] to compute the flux vertically averaged along the wind-exposed part (h) of the sensor (of full height H):  $F_{b,corrected} = F_{b,measured} \times H/h$ . All the figures and tables have been modified accordingly. A new paragraph has also been added in Sect. 3.1.3 to explain the correction:

Throughout the year, the lowermost FlowCaptTM gets partially buried due to snow accumulation. At D47, a SR50 acoustic depth sensor monitored the surface elevation continuously between 2010 and 2012 showing that the wind-exposed part of the H=1 m high sensor was  $h\approx 0.6$  m in 2011. Building from Amory et al. [3], the measured flux has therefore been scaled at each time step by H/h to obtain the particle mass flux vertically averaged over the wind-exposed part of the sensor, consistently with the sensor calibration principle which implicitly assumes integration over its full exposed height H, requiring correction when only a fraction h is exposed. At D17, the SR50 sensor was deployed in December 2012, thus after the 2011 analysis period considered here. No correction can therefore be applied for this station which likely results in a underestimation of the flux magnitude. As the D17 instruments are raised back manually to original heights at the beginning of each summer field campaign, the underestimation is likely more important during the winter and spring season but this cannot be properly quantified.

(c) Recently, Nishimura et al. (2024) published almost three months of blowing snow profile measurements at Mizuho Station, East Antarctica, which had been partly analyzed in Nishimura and Nemoto (2005). The profile was measured using four snow particle counters. As the uppermost sensor was at a height of 9.6 m, this dataset offers an excellent opportunity to evaluate more directly the modeled mass flux at the first grid level.

We thank the reviewer for pointing us to this dataset we were not aware of. Indeed, it seems extremely relevant to evaluate meteorogical variables and blowing snow fluxes at heights corresponding to typical GCM first model levels. We have added the reference in the last section of the paper when acknowledging the value of data collected along meteorological masts in Antarctica.

I recommend major revisions to take these suggestions into account and address the following comments. With revisions, the study has the potential to become a valuable contribution to the research field.

**Specific comments**

(3) l. 41-45: It would be worth to mention the paper of Saigger et al. (2024), which describes an intermediate-complexity parameterization of blowing snow in the WRF model. Please also provide examples on how an intermediate-complexity parameterization differs from more complex ones.

This is an important point that indeed deserves clarification in the manuscript. We use the 'intermediate complexity' terminology to emphasize that our blowing snow scheme does not rely on a sophisticated surface snow scheme that explicitly accounts for densification effects associated with snow erosion (such

DATA SET: D47\_halfh\_2010\_2012

actual top height above ground of the lower FlowCapt sensor

Figure 1: Time series of the top height of the lower FlowCapt sensor at D17 (top panel) and D47 (bottom panel) stations as provided by the Amory [2]'s dataset. This height is estimated thanks to a SR50 acoustic depth sensor, except in 2010, 2011 and 2012 at D17 where the reported value corresponds to the most recent visual inspection (only during the summer season) of the station.

as SNOWPACK in CRYOWRF for instance). Moreover, we want to stress that we consider a relatively simple one-moment treatment for the blowing snow water species (unlike a 2-moment treatment in CRYOWRF and Méso-NH) and that one scheme does not include an additional vertical discretization of the surface layer such as that in CRYOWRF and Méso-NH). We have modified the parameterization introduction paragraph and added a reference to Saigger et al as follows: We therefore follow an intermediate-complexity approach in the sense that the parameterisation does not require a very sophisticated snow scheme - such as SNOWPACK for CRYOWRF for instance [12] - and does not include an additional discretization of the surface layer such as in Vionnet et al. [16]. Such as in MAR [4], RACMO [6] and WRF [11], a blowing snow flux is directly calculated between a fully parameterised saltation layer near the surface and the first model level at a few meters above the ground surface. However, the specific content of blowing snow particles in suspension  $q_b$  (in kg kg-1) is

treated as an independent water variable in the model - unlike in MAR for instance - to properly distinguish the blowing snow contribution to precipitation and radiative effects from that of typical clouds.  $q_b$  is advected by the dynamical core and vertically transported by turbulent diffusion. However, we keep a one-moment treatment for the blowing snow water species and does not consider an additional prognostic estimation of the number of blowing snow particles [16, 12].

(4) l. 101: The vertical transport of blowing snow does not only occur through turbulent diffusion but also sedimentation. Although the individual terms of the prognostic equation for blowing snow are presented later, I propose to add the prognostic equation for qb (combining the left-hand-sides of Eqs. 6, 8, 13 and the advection term) to better guide the reader.

This is a very good idea. We have added the full evolution equation at the end of the introduction paragraph of Sect. 2.2 (general concepts of the parameterization) and added the word 'sedimentation' referring to the vertical transport.

(5) l. 113: Should  $(\rho_i/\rho_{s0}-\rho_i/\rho_s)$  be an exponent as in Amory et al. (2021)? Otherwise, the threshold friction velocity is not  $u_{*t0}$  but zero for new snow  $(\rho_s = \rho_{s0})$ .

The exponential was missing in the equation (with the density term in exponent), thank you very much for pointing this mistake. The equation has been corrected. The corresponding code was checked and was correct.

- (6) l. 143: Is an exponent missing in Eq. 3 and is the associated citation of Pomeroy (1989) correct? Both is inconsistent with the corresponding description of MAR below Eq. 5 in Amory et al (2021).
- The 1.27 exponent was missing in the equation, thank you for noticing. It was corrected. The corresponding code was checked and was correct.
- (7) l. 179: 'radius  $r_b$ ': Do you assume a constant particle radius at all heights? This should be clearly stated and the value of the radius should be specified.

Thank you for pointing this shortcoming. We indeed assumed a monodisperse population of blowing snow particle with constant particle radius (so constant also with height). We deliberately do not use an empirical law for the decrease of  $r_b$  with z (such as as in Saigger et al. [11]) as this dependency is very context-dependent. Developing a new version of the parameterization with a full 2-moment treatment of a non-monodisperse particle size distribution is an avenue for improvement. In the current version of the parameterization,  $r_b$  is thus tuning parameter, whose value can be changed in the namelist file and that is equal to  $50 \mu m$  by default. This is now specified in the manuscript in Sect. 2.5. Following your comment we have implemented the height-dependent radius of Saigger et al. [11] as an option in our model, but testing it is left for future work.

(8) l. 180: In the cited publications, I could not find Eq. 8 but only similar

equations. When trying to derive Eq. 8, I arrived at a slightly different equation. In contrast to Rutledge and Hobbs (1983), the ventilation factor seems to be missing; or is it included in the terms A' and B'? Where does pi in the denominator come from? I assume that the final units should be kg kg-1 s-1; is the air density in the numerator needed?

Thank you very much for the very careful proof reading of the equations. Indeed some mistakes were present ( $\rho$  and  $\pi$  should not be there) due to an insufficient care taken when re-copying the equations in the manuscript. The equations have been corrected in the manuscript. Note that the code was based on the correct version of the equations and is correct. Please also note that we do not consider a ventilation factor here due to the relative small fall velocity of blowing snow particles (as in Muench and Lohmann [10] for ice crystals) and because the fall velocity (upon which the ventilation factor is based) is a tuning parameter that we want to control the sedimentation process only.

- (9) l. 197: When rearranging Eq. 10 to obtain Eq. 11, did you forget the fraction  $6\rho/(\rho_b pir_b^2(A'+B'))$ ? Thank you very much for noticing this mistake. It has been corrected. The corresponding code has been verified and it is correct.
- (10) l. 207 209: How does the time scale  $\tau_m$  affect the simulation? As this time scale is 10 min or lower and the typical time step is 15 min, the blowing snow particles melt and evaporate within one time step in the considered situation. Does  $\tau_m$  influence the radiative effect of blowing snow in the time step? This is a good point but as you notice, the melting time scale is close to the time step value. Subsequently, blowing snow particles melt in a few time steps when they enter a layer with positive Celcius temperature. We could have chosen to make them instantaneously melt, but the continuous formulation is preferable especially for numerical aspects. No specific sensitivity test was performed to assess the effect of  $\tau_m$  upon the radiative effect of blowing snow. In fact, we can reasonably expect no major impact as small blowing snow particles do not survive more than a few minutes at positive temperatures.
- (11) l. 265: Can you describe the terrain surrounding the measurement sites? This might be relevant for the comparison of modeled and measured wind speeds.

Following your comments, we have added a paragraph in Section 3.1.2:

'The topographic channelling of the gravity-driven near-surface flow gives coastal Adélie Land the most intense sustained surface winds on Earth [Parish'2006] and very frequent and intense blowing snow events [2, 15]. This region consists of a sloping snowfield with no major relief but a break in slope at nearly 210 km inland at about 2100 m a.s.l., downstream of which D47 and D17 are located.'

(12) l. 286 - 288: The sentence sounds like you use a weighted average to combine observed and modeled SMB into a final estimate. After checking Agosta et al. (2019), however, I assume that you perform a weighted average

of SMB observations that fall into the same ICOLMDZ grid cell. Please clarify. Apart from that, if an observation spans a longer period than the 5-year simulation period, do you only use the observations during the simulation period or the whole observation period?

Our sentence is misleading and it has been clarified as follows:

We then perform a weighted average – by weighting with the observed accumulation duration – of SMB observations that fall into the same ICOLMDZ grid cell, as in Agosta et al. [1]

Moreover, only observations during the simulation period have been used. This is now clarified in the manuscript in Sect. 3.1.2.

(13) l. 292: 'simulated roughness length': Prescribed roughness length would be a better wording if it is a constant value for snow and ice surfaces as stated in l. 70.

Following your recommendation, we have changed 'simulated roughness length' with 'prescribed roughness length value in the model'.

(14) l. 367 – 369: For D47, the agreement between modeled and measured RH (Fig. 3f) is not discussed. It should at least be mentioned that lower RH values in the model compared to the measurements can be expected at D47 as the first model level is above the measurement height at this site.

This is indeed something that has to be added. Following your recommendation, the following sentence has been inserted in the main text:

At D47, part of the overall low RH bias can be explained by the difference between the height of the first model level and that of the measurement ( $\approx 2.2 \text{ m}$ ).

(15) l. 397: 'the simulated frequency is more realistic in July, August and December at D47': In July and August, this may, at least partly, be due to an overestimation of wind speed.

We partly agree. Indeed the wind speed is overestimated during the extended winter at D47, but not only in July and August. It is thus not clear to what extent this wind speed overestimation explains the better frequency during these months. We therefore prefer not modifying the text.

- (16) l. 412: 'turbulent latent heat flux': I assume this refers only to the flux at the surface and does not include blowing snow sublimation, right? Yes indeed, this flux only accounts for the sublimation of surface snow. This is now specified in the text.
- (17) l. 423: 'a few tens of K' should be a few tenths of K if you mean a fraction of 1 K (?)

Yes of course, thank you for pointing this mistake out, it has been corrected.

(18) l. 432-434: Does the blowing snow parameterization lead to an improved agreement with the SMB measurements? Or is a direct comparison not meaningful?

This is a delicate point. In fact, the direct model-observation comparison does not show a clear improvement (see Figure 2) when including the blowing snow parameterization, with very close mean bias and RMSE (not shown). Nonetheless, almost all the SMB observations correspond to locations where the simulated SMB is not very affected by blowing snow (see Figure 10b in the main paper). The present comparison is therefore two limited to robustly conclude on the possible beneficial (or detrimental) impact of blowing snow on the Antarctic SMB in global simulation. We have added the following sentence in the manuscript:

The difference can locally reach several tens of kg m-2 yr-1 but the absence of SMB measurements in the regions with the strongest changes prevents us from concluding about a possible improvement or deterioration of the local SMB modelling.

Figure 2: Scatter plot of simulated versus observed SMB during the 5-y simulation period. Left (resp. right) panel shows the simulation without (resp. with) blowing snow.

(19) l. 442: 'no overall increase in large-scale precipitation': Do you expect that large-scale precipitation would increase if blowing snow particles were considered as ice-nucleating particles in cloud formation?

This is a very good point and unfortunately we have no clear answer to this question. One may indeed suggest that seeding mixed-phase clouds with blowing snow crystals would enhance ice precipitation production. Unfortunately, the current version of LMDZ does not include a direct interaction between the blowing snow scheme and the cloud microphysics scheme. In other words, blowing snow crystals cannot serve as INP and do not interact with clouds such as through the Wegener-Feidensen-Bergeron and riming processes (see a thorough description of mixed-phase clouds representation in LMDZ in Raillard et al. 2025, DOI: 10.22541/essoar.175096287.71557703/v1). Such a discussion is deemed out of the scope of the present paper and we prefer not adding an additional paragraph to discuss this specific aspect.

(20) l. 456: 'the moistening effect of the surface layer is underestimated': This conclusion is not sufficiently discussed. You mention in l.368 that 'the model fails to capture periods of saturation [at D17]' but is it clear that this is due to an underestimation of blowing snow sublimation? Or could there be other reasons such as an overestimation of air temperature?

We agree, and this is a delicate point. We certainly underestimate the blowing snow sublimation but is this due to an overall underestimation of the blowing snow concentration near the surface (but we have no reliable quantitative reference) or is this more attributable to a inadequate parameterization of the sublimation (issue related to calibration or to the underlying hypothesis regarding the particle size distribution ...)? The sensitivity to the  $\gamma_{sub}$  was assessed and it is in fact possible to increase the near-surface relative humidity with higher  $\gamma_{sub}$  values but this deteriorates the statistics of blowing snow events' detection (due to overly long residence time of blowing snow particles). We now therefore specify (in the main text) that this bias is likely due to an underestimated sublimation.

(21) l. 457: 'During winter, wind speed, snow flux amplitude and occurrences at D47 are well simulated': This statement contradicts l. 394-395: 'the July value [of the blowing snow flux intensity] - very close to the FlowCaptTM measurements between 0 and 1 m - is likely overly strong'.

Thank you, we agree, this statement was not fully consistent with the Results section. It has been reformulated as: During winter at D47, the monthly mean wind speed is overestimated by about 1 m s-1 and a mean cold bias ranging between 1 and 2 K is noticeable. The snow flux occurrence fits well the observations in July and August but the amplitude is probably overestimated.

**Technical corrections**

 $\left(22\right)$  Typo in short summary: 'Simulations avec evaluated using measurements in Antarctica.'

Corrected.

(23) l. 99: 'specific content of blowing snow particles in suspension qb': I suggest to provide units (kg kg-1) or call it the mass mixing ratio to be more precise.

It is a specific content, namely a mass of blowing snow per mass of humid air, not a mixing ratio. We now specify the units kg  $kg^{-1}$ .

- (24) l. 113: Equation number is missing. Thank you, this has been corrected.
- (25) Caption of Fig. 6: averahed should be averaged. Corrected.

- (26) l. 151: U should be defined in the text below Eq. 5. We now specify 'U the wind speed at the first model level'
- (27) Caption of Fig. 1 (4th line): T, P, and  $RH_i$  should be defined here as they have not been defined in the main text yet. A full stop is missing after 'converge'.

The caption has been modified accordingly.

(28) l. 205: The 'reference curve' should also be explained in the main text, not only in the figure caption.

We snow specify 'the reference curve corresponding to the solution with a 1 s time step.'

(29) l. 208: I assume that T in Eq. 12 is air temperature but it is not defined in the text.

Thank you, 'T' is now defined in the main text.

- (30) Figure 5: Black and blue dots are difficult to distinguish. Can you use different colours or show separate plots for model and measurement results? Figure 5 has been modified using different colors and symbols to make it clearer and facilitate the distinction between model and measurements results.
- (31)l. 397: 'despite but underestimated': Please check meaning and grammar.

Thank you. The end of the sentence now reads 'while the simulated frequency is more realistic in July, August and December at D47 but is underestimated the rest of the year.'

(32) Caption of Fig. 8: 'Net surface radiative flux' should be net shortwave surface radiative flux. The dash-dotted line in panel e should be explained. Please mention the sign convention for the surface turbulent fluxes (positive = downward?).

The caption of the figure has been corrected following the three recommendations.

- (33) l. 422: a reference to Fig. 9b is missing. The reference to Fig. 9b has been added, thank you for noticing.
- (34) l. 480 483: Please check the grammar. The sentence has been rephrased as follows: 'Then, we expressed the concentration of particles in the saltation layer  $q_{b,salt}$  using a formula from the saltation model of Pomeroy (1989) in which the particle mass flux in the saltation layer is assumed uniform in height. Such a model is in contradiction with the well-documented exponential decay of the particle mass flux.'
  - (35) l. 671: There is a typo: 'dOI:'.

**References**

- [1] C. Agosta et al. "Estimation of the Antarctic surface mass balance using the regional climate model MAR (1979–2015) and identification of dominant processes". In: *The Cryosphere* 13.1 (2019), pp. 281–296. DOI: 10.5194/tc-13-281-2019.
- [2] C. Amory. "Drifting-snow statistics from multiple-year autonomous measurements in Adélie Land, East Antarctica". In: *The Cryosphere* 14.5 (2020), pp. 1713-1725. DOI: 10.5194/tc-14-1713-2020. URL: https://tc.copernicus.org/articles/14/1713/2020/.
- [3] C. Amory et al. "Performance of MAR (v3.11) in simulating the drifting-snow climate and surface mass balance of Adélie Land, East Antarctica". In: Geoscientific Model Development 14.6 (2021), pp. 3487-3510. DOI: 10.5194/gmd-14-3487-2021. URL: https://gmd.copernicus.org/articles/14/3487/2021/.
- [4] H. Gallée, G. Guyomarc'h, and E. Brun. "Impact of snow drift on the Antarctic Ice Sheet surface mass balance. Possible sensitivity to snow surface properties". In: *Boundary-Layer Meteorol* 99 (2001), pp. 1–19.
- [5] Mark Gordon, Sergiy Savelyev, and Peter A. Taylor. "Measurements of blowing snow, part II: Mass and number density profiles and saltation height at Franklin Bay, NWT, Canada". In: Cold Regions Science and Technology 55.1 (2009), pp. 75-85. ISSN: 0165-232X. DOI: 10.1016/j. coldregions.2008.07.001. URL: https://www.sciencedirect.com/ science/article/pii/S0165232X0800102X.
- [6] J. T. M. Lenaerts et al. "Modeling drifting snow in Antarctica with a regional climate model: 1. Methods and model evaluation". In: Journal of Geophysical Research: Atmospheres 117.D5 (2012). D05108. DOI: 10. 1029/2011JD016145.
- [7] G. W. Mann, P. S. Anderson, and S. D. Mobbs. "Profile measurements of blowing snow at Halley, Antarctica". In: *Journal of Geophysical Research:* Atmospheres 105.D19 (2000), pp. 24491–24508. DOI: https://doi.org/ 10.1029/2000JD900247.
- [8] Raleigh L. Martin and Jasper F. Kok. "Wind-invariant saltation heights imply linear scaling of aeolian saltation flux with shear stress". In: Science Advances 3.6 (2017), e1602569. DOI: 10.1126/sciadv.1602569. eprint: https://www.science.org/doi/pdf/10.1126/sciadv.1602569. URL: https://www.science.org/doi/abs/10.1126/sciadv.1602569.

- [9] D. B. Melo, A. Sigmund, and M. Lehning. "Understanding snow saltation parameterizations: lessons from theory, experiments and numerical simulations". In: *The Cryosphere* 18.3 (2024), pp. 1287–1313. DOI: 10.5194/tc-18-1287-2024. URL: https://tc.copernicus.org/articles/18/1287/ 2024/.
- [10] Steffen Muench and Ulrike Lohmann. "Developing a Cloud Scheme With Prognostic Cloud Fraction and Two Moment Microphysics for ECHAM-HAM". In: Journal of Advances in Modeling Earth Systems 12.8 (2020), e2019MS001824. DOI: https://doi.org/10.1029/2019MS001824.
- [11] Manuel Saigger et al. "A Drifting and Blowing Snow Scheme in the Weather Research and Forecasting Model". In: *Journal of Advances in Modeling Earth Systems* 16.6 (2024), e2023MS004007. DOI: https://doi.org/10.1029/2023MS004007.
- [12] V. Sharma, F. Gerber, and M. Lehning. "Introducing CRYOWRF v1.0: multiscale atmospheric flow simulations with advanced snow cover modelling". In: Geoscientific Model Development 16.2 (2023), pp. 719-749. DOI: 10.5194/gmd-16-719-2023. URL: https://gmd.copernicus.org/articles/16/719/2023/.
- [13] A. Sigmund et al. "Parameterizing Snow Sublimation in Conditions of Drifting and Blowing Snow". In: *Journal of Advances in Modeling Earth Systems* 17.5 (2025), e2024MS004332. DOI: 10.1029/2024MS004332.
- [14] E. Vignon et al. "Designing a Fully-Tunable and Versatile TKE-l Turbulence Parameterization for the Simulation of Stable Boundary Layers". In: *Journal of Advances in Modeling Earth Systems* 16.10 (2024), e2024MS004400. DOI: https://doi.org/10.1029/2024MS004400.
- [15] Etienne Vignon et al. "Gravity Wave Excitation during the Coastal Transition of an Extreme Katabatic Flow in Antarctica". In: *Journal of the Atmospheric Sciences* 77.4 (2020), pp. 1295–1312. DOI: 10.1175/JAS-D-19-0264.1.
- [16] V. Vionnet et al. "Simulation of wind-induced snow transport and sub-limation in alpine terrain using a fully coupled snowpack/atmosphere model". In: *The Cryosphere* 8.2 (2014), pp. 395–415. DOI: 10.5194/tc-8-395-2014. URL: https://tc.copernicus.org/articles/8/395/2014/.

---

## Author Comment (AC2)

**Revision of**

**'Intermediate-complexity Parameterisation of Blowing Snow in the ICOLMDZ AGCM: development and first applications in Antarctica'**

Etienne Vignon, Nicolas Chiabrando et al.

October 30, 2025

This document contains the response to a review of 'Intermediate-complexity Parameterisation of Blowing Snow in the ICOLMDZ AGCM: development and first applications in Antarctica' submitted to EGUSPHERE for possible publication in Geoscientific Model Development. Comments from the Reviewer are in black and answers are in blue. Paragraphs that have been added or modified during the revision process are copied in purple.

**Reviewer #2**

The following is a review of "Intermediate-complexity Parameterisation of Blowing Snow in the ICOLMDZ AGCM: development and first applications in Antarctica" By Étienne Vignon and others.

This manuscript describes the integration and evaluation of a blowing snow parameterization for Antarctica. Blowing and drifting snow on the surface of ice sheets, particularly Antarctica, has been shown to be a nontrivial contribution to surface mass balance. However, representation of this process is included in few regional-scale models used to estimate ice sheet surface mass balance. This study is novel in that it investigates the utility and computational burden of including an intermediately complex parameterization of blowing and drifting snow into an atmospheric general circulation model (GCM) that has been recently modified to better capture near-surface winds. The authors present the model design and implementation, model evaluation, and impact on surface mass balance including discussion on thermodynamic and cloud effects due to the new model capabilities. Estimates of blowing snow show skill against observations in the test region of Adélie Land and are comparable to results from a regional climate model. Finally, the authors present results of global-scale simulations with and without blowing snow and show general climatological agreement with observations with respect to surface mass balance.

Overall, I find that the manuscript is organized and nicely written. Model assumptions are clearly articulated within the text. For the most part, I find the modeling procedure easy to follow and that the figures are of good quality. The paper focuses on describing the model and evaluation against observations, which is appropriate content for GMD. As a result, I am recommending it for publication after suggested edits.

We are grateful to the referee for the careful and thorough review of our manuscript. We sincerely appreciate all the comments, which have significantly helped us improve the study. Please find below our responses to each comment.

Specifically, I would like to see the authors expand the discussion to explicitly provide closing thoughts about some of the key motivating questions that are brought up in the paper introduction, specifically those related to whether blowing snow should be included in GCM's. These are important questions that are touched upon early in the manuscript that make this study particularly engaging to the audience, and I think it would improve the paper to touch upon them again after the results are presented. Please see more detailed comments below.

**Questions and suggestions**

Line 27 and Line 447: I agree, the past research and this study raises this important question. It would really help round the paper out if the authors explicitly gave an opinion of the answer to this with respect to their results and what is presented in the discussion. In my view, the statement does not have to be strongly conclusive of final in any way (considering the uncertainties that are discussed) but since the important questions is raised, it would strengthen the paper to have it addressed directly within the text.

Thank you for this comment which invites us be more explicit regarding our opinion. The overall non-negligible impact of blowing snow on the SMB and on the coastal surface energy budget, jointly with the very weak impact on the other climate variables at lower latitudes are strong arguments in favour of including blowing snow in global climate model simulations, in particular in models used for polar-oriented studies and those which are intended to be coupled with ice sheet model (e.g., [5]). This statement should however be mitigated. First, our study has not assessed – and thus demonstrated – a possible improvement in terms of simulated radiative fluxes near the Antarctic coast and of the SMB with respect to observations, which calls for further evaluation work. Second, the inclusion of blowing snow has a non negligible additional computational cost  $(\approx 4\%)$  which might be a limitation when running the model over very long period of time or when carrying out ensemble experiments. Our opinion is thus the following. As it does not affect the global climate properties but add some sophistication in terms of process representation on the ice sheets, we recommend the inclusion and use of blowing snow parameterizations in global models for specific runs of particular interest for polar studies. However, further evaluation - in terms of SMB and radiative fluxes in particular - is needed to confirm that the sophistication provided by the present scheme in ICOLMDZ goes along with an improvement of the model at the Antarctic scale. The following paragraph has been added in the discussion:

The overall significant impact of blowing snow on simulated surface mass balance (SMB) and the coastal surface energy budget, combined with its very limited effect on the climate at lower latitudes, are strong arguments in favor of including blowing snow processes in global climate model simulations—particularly in models designed for polar-focused studies, such as those coupled with ice sheet models (e.g., [5]). However, this statement should be nuanced. First, our study has not demonstrated a clear improvement in simulated radiative fluxes and SMB when compared with observations, highlighting the need for further evaluation at the Antarctic scale. Second, including blowing snow introduces a small but non-negligible additional computational cost, which may become a limiting factor for long-term simulations or ensemble experiments, especially when increasing model resolution to better capture the spatial variability of precipitation over the ice sheets. Therefore, the use of blowing snow parameterizations in global climate models can be recommended for simulations specifically targeting polar processes. Nonetheless, further evaluation is required to confirm that the added complexity of the current scheme in ICOLMDZ results in an overall improvement of model performance at the Antarctic scale.

Line 33 and Line 446: It is clearly noted that transport of mass off the continent is an important part of the quantification of SMB by the model, and that including wind-blown snow could represent this discrepancy. Including some statistics about how much snow is estimated to be transported off the continent in the global runs would be very helpful for the reader to grasp if the process is significant to the GCM simulations. One suggestion is to make direct comparison with these estimates of percent change from other studies that are referenced in the paper, to offer insight into how important the process is in the GCM. (This suggestion ties in strongly with the above L27 comment).

We completely agree that adding such an information in the paper with a comparison with previous estimates in the literature would be very informative. However, properly estimating the quantity of blowing snow advected outside of the continent requires computing the flux perpendicular to the coastline at each time step. Unfortunately, our output files do not have the necessary time resolution to perform the calculation. Therefore, we would have to reconfigure the content and resolution of our output files and re-run the global simulations which are numerically costly. This reason explains what we can unfortunately not satisfy your request here.

Line 122: Does snowfall accumulated here also include snow that is deposited (sedimentation?). It is unclear from the text if the snow being deposited is feeding back to this aging estimation. Perhaps a rephrasing of this paragraph and specifying what is meant by snowfall accumulation would help with the confusion.

This is a very good point. The aging parameterisation here aims to account for the aging since the last 'fresh snow' precipitation, thus excluding blowing snow sedimentation. This is now specified in the main text.

Line 125: Is the snow age "reset" from a different value when it snows? Or is it just "set" to 0 for new snow when it snows? This wording is a bit confusing, especially since the statement above suggests that if snow falls and it happens to be eroded then the densification equation is used. But in this specific case, would the value be 0 even though it snowed but did not actually accumulate? This might just be a question of the terms used for the different ways snow can accumulate, and I suggest using precise wording and definition for each. As noted above, maybe a rephrasing of the entire paragraph would help.

Thank you for this comment. Indeed, the snow age is reset to 0 as soon as fresh snow accumulates i.e. if there is sufficient snowfall such as some fresh snow remains after the erosion process during the time step. This point is now clarified in the paragraph and the wording of the previous paragraph has been adjusted for better consistency.

'The snow age is reset to 0 as soon as some fresh snow accumulates during the time step that is, if some fresh snow corresponding to the snow that falls at the given time step remains after the erosion process.'

Lines 126-129: This last sentence is also awkward and difficult for the reader to follow.

We have reformulated as follows:

'Within each time step  $\Delta t$ , we do not a priori know the time that corresponds to the erosion of the superficial fresh surface snow - which is the snow that has fallen during the time step - and the time that corresponds to the erosion of the underlying, and thus older, snow layers. We thus assume that the fresh snow erosion occurs during a fraction  $\omega_f$  of  $\Delta t$  that depends on the relative difference between the fresh snow erosion flux Er and the snowfall during the time step  $Sf\colon \omega_f = e^{-(\frac{|Er-Sf|}{Sf})}$ ,

Line 211: Similar to the above questions, does this end up getting treated the same as precipitation in some way? What is "precipitation" consisting of? How does deposited snow feed back into the densification equation?

The sedimentation of blowing snow particles is indeed treated the same way as the sedimentation of precipitation hydrometeors such as snowflakes. Your comment make us realize we used independently 'sedimentation' and 'precipitation' of blowing snow, which confuse the reader. The text has been corrected to use the wording 'sedimentation' only. The deposited blowing feeds back into the densification equation through the first exponential term of equation 3.

Line 352: Could you add a comment on if we should expect that the parameterization significantly affects wind or temperature? Is this surprising at all? Blowing snow is expected to have an impact on near-surface temperature through

latent effects (sublimation) and radiative effects. This is in fact what is shown and discussed in Figure 8 in the main paper. The effect on the wind speed is very weak, more subtle and indirect. The modulation of near surface temperature and stability can affect the katabatic forcing [2], and the effect of blowing snow particles on atmospheric stability can modulate the near-surface turbulence and drag, with very little impact on wind speed [1], but this effect is not taken into account in our parameterization. Overall, blowing snow has a very little effect on wind speed. Coming back to your comment on line 352 where we discuss the summertime time series, we overall expect little effect of blowing snow on temperature as the wind speed and blowing snow fluxes have a moderate magnitude. In the main text, we have modified the corresponding paragraph as follows:

The activation of the blowing snow parameterisation has overall a little effect upon simulated wind and temperature time series. In fact, the moderate blowing snow fluxes and concentrations in January are not sufficiently strong to significantly affect the air temperature and atmospheric stability - and subsequent katabatic forcing - through particles sublimation.

Lines 391-392: Figure 6e does not appear to show that there is an over-estimation of flux during these months. Perhaps I am misunderstanding the comment and if so, please rephrase.

The sentence has been rephrased as follows:

The magnitude of the simulated blowing snow flux at the first model level at D17 is either close to or even exceeds the FlowCaptTM measurements between 0 and 1 m (Figure 6e) and is therefore likely overestimated, concurring with the too strong simulated wind speeds at this station particularly during the extended winter.

Lines 431-432: Do you think this is associated with the spatial resolution that you needed to run with? Or do you think there is physics that is missing to capture these winds? Please add a brief comment to this effect in the text. The simulation of the Foehn effect over the Antarctic Peninsula is very sensitive to the model resolution [4], so are the topography-induced circulations. At this stage, we cannot speculate on possible issues regarding the physics content for the representation of Antarctic Peninsula Foehn winds but we can add that the resolution we are using in the global runs is insufficient to finely capture the relief contrast and associated winds. The text has been modified as follows: Comparison with observations (circles) reveals a reasonable agreement, except to the east of the Peninsula. This might be attributed to an excess of precipitation associated with a possible underestimated Foehn effect due to the quite coarse horizontal resolution employed in the global runs.

Line 434: Here you state that the differences should be considered negligible locally. In the same vein as my earlier questions about the significance of blowing snow and its continental-scale magnitude, I suggest the authors bring the implications of this result back up in the discussion. It seems to be an important conclusion to this work. Is this value negligible locally but more significant continentally?

Our statement was not very clear because we wanted to emphasize that the differences are not negligible locally as their magnitude can exceed  $10 \text{ kg m}^{-2} \text{ yr}^{-1}$ . Following your recommendation, we have added the following sentence in the Discussion section:

The difference is locally not negligible as it can exceed several tens of kg  $\,\mathrm{m}^{-2}\,\,\mathrm{yr}^{-1}$  in magnitude..

Figure 10: Caption – please specify in the caption the difference between "observed SMB values" in a) and SMB observations in b). My understanding from the text is that they are different because the grey dots in a) are the points of the observations themselves and the circles in a) are those positions interpolated onto the GCM grid. This would make sense why there are many more circles in a) closer to the pole. But it seems like there still should be way more circles in a) in Thwaites and Ross area. Are the grey also showing the locations of observations that do not meet the criteria of use? If so, those observation locations should probably be removed from b). It is also unclear why there are values off the coast of Ronne when there are no grey dots in b) near those locations. Please clarify this in the text and in the caption.

Thank you for pointing this incomplete explanation. Circles in panel a show the the observed accumulation data averaged onto LMDZ grid cells and weighted by the time-length of the corresponding observation during the considered period (here 2000-2005). Grey points in panel b show the locations of all SMB observations. This aspect is now clarified in the figure's caption:

'Grey dots in panel (b) show the location of all SMB observations available in the observation dataset. Circles in panel a are the averaged values from observations within each model grid cells, the average being calculated by weighting with the observed accumulation duration. Let's recall that we discard observations covering less than 3 years and only keep observations during the 5-year simulation period.'

Figure 10 and Line 440: The plots here are a little confusing, because there are many SMB values outside of the continent. I realize that is might be where the blowing snow is depositing, but for this case in b), is precipitation outside of the ice sheet being considered for both the with and without blowing snow simulations? (I suspect maybe yes since there are negative values for b) outside of the ice sheet.) Or are the values outside of the ice sheet only AIS-sourced values (i.e. wind-blown and not atmospheric precipitation). In that case, can the values outside really be considered true SMB? I guess it is also possible that the GCM ice sheet grid extends past the black coastline boundaries drawn on the figure. Please try to revise the text and caption to be clearer about what is being shown.

Thank you for pointing this issue out. In fact, a very subtle mistake was present in our interpolation script (affecting the interpolation in longitudes leading to inconsistencies in the treatment of coastal grid points). It has been fixed. The figures have been corrected and are now shown in the revised version of the manuscript.

Figure 11: I have a similar confusion to the above, over Fig. 11 which shows (precipitation – erosion). What is "precipitation" in this context? Presumably it is the blowing snow deposition? Most likely, clarifying the text and caption for each figure would alleviate most of the confusion.

Indeed we meant 'blowing snow deposition'. The text and caption have been modified accordingly.

**Minor edits**

Line 45: This statement is awkward, please rephrase. Maybe use "constraints" instead of "constrains"?

Thank you, we have rephrased as follows:

Several parameterizations of snow erosion and transport have been proposed so far. However, to our knowledge, all of them were developed for mesoscale models and often involve a level of complexity — as well as an additional computational cost, particularly due to the inclusion of extra water species — that is not always compatible with the constraints of global climate simulations.

```
Line 104: "in" \rightarrow of Corrected.
```

Line 159: "authors"  $\rightarrow$  authors' Corrected.

Lines 306-307: precipitation "is" diagnosed? "prevent" -; "prevents us"? In general, this sentence is awkward. Please rephrase for clarity.

The sentence has been rephrased as follows:

However, the LMDZ cloud scheme diagnoses the vertical snowfall flux at each time step but does not compute the specific content – or mass mixing ratio – of snow particles [3]. This prevents us from robustly estimating a horizontal flux of all the particle categories – including snowflakes – from model outputs.

```
Line 313: "follows" Corrected.

Line 317: "measures" \rightarrow measurements (?) Corrected.

Line 323: "event" \rightarrow events Corrected.

Line 344: "month" \rightarrow the month
```

Corrected.

```
Line 393: "at" \rightarrow during (?) Corrected.
```

Line 423: Should this be "tenth of K"? A few tens of K seems very large. Yes of course, this has been corrected.

Line 424: Antarctic Corrected.

**References**

- [1] H. Gallée, G. Guyomarc'h, and E. Brun. "Impact of snow drift on the Antarctic Ice Sheet surface mass balance. Possible sensitivity to snow surface properties". In: *Boundary-Layer Meteorol* 99 (2001), pp. 1–19.
- [2] Y. Kodama, G. Wendler, and J. Gosink. "The effect of blowing snow on katabatic winds in Antarctica". In: *Ann Glaciol* 6 (1985), pp. 59–62.
- [3] J.-B. Madeleine et al. "Improved representation of clouds in the LMDZ6A Global Climate Model". In: J Adv Model Earth Sys 12 (2020). DOI: 10. 1029/2020MS002046.
- [4] Andrew Orr et al. "Comparison of kilometre and sub-kilometre scale simulations of a foehn wind event over the Larsen C Ice Shelf, Antarctic Peninsula using the Met Office Unified Model (MetUM)". In: Quarterly Journal of the Royal Meteorological Society 147.739 (2021), pp. 3472–3492. DOI: 10.1002/qj.4138.
- [5] Robin S. Smith et al. "Coupling the U.K. Earth System Model to Dynamic Models of the Greenland and Antarctic Ice Sheets". In: Journal of Advances in Modeling Earth Systems 13.10 (2021), e2021MS002520. DOI: 10.1029/2021MS002520. URL: https://agupubs.onlinelibrary.wiley.com/doi/abs/10.1029/2021MS002520.

---

## Author Comment (AC3)

**Revision of**

**'Intermediate-complexity Parameterisation of Blowing Snow in the ICOLMDZ AGCM: development and first applications in Antarctica'**

Etienne Vignon, Nicolas Chiabrando et al.

October 30, 2025

This document contains the response to a review of 'Intermediate-complexity Parameterisation of Blowing Snow in the ICOLMDZ AGCM: development and first applications in Antarctica' submitted to EGUSPHERE for possible publication in Geoscientific Model Development. Comments from the Reviewer are in black and answers are in blue. Paragraphs that have been added or modified during the revision process are copied in purple.

**Reviewer #3**

This study aims to incorporate blowing snow physics in ICOLMDZ global climate model to improve the representation of Antarctic SMB. The manuscript is well-written and structured. The study is appropriate and fits within the scope of GMD and timely. However, the main novelty of the paper wrt 'intermediate-complexity' blowing snow parameterisation for a GCM needs more explanation, justification, and rewriting. The authors mention that increasing the grid resolution near the surface would 'unreasonably increase' the computational cost, however contrary to the approach used in typical RCMs, it appears that in the current approach the authors also run the blowing snow model at atmospheric heights (all model levels) where there would be no blowing-snow, which is also computationally not efficient. In addition, it appears there are few inconsistencies wrt to the parameterisations and observations, which need to be addressed before the manuscript is accepted for publication.

We sincerely thank the referee for their careful and constructive review of our manuscript. Please find below our responses to each comment, along with explanations of the revisions that have genuinely helped improve the paper.

The manuscript would also benefit from proof-reading. There are multiple typographical errors in very important places, which makes it a little difficult to read.

A careful proof-reading was performed and we hope no typographical errors remains in the revised version of the manuscript.

**Major comments**

1. Line 100: It is not clear to me what exactly author's mean by intermediate-complexity, please elaborate in comparison with other implementations in mesoscale models.

This is an important point that indeed deserves clarification in the manuscript. We use the 'intermediate complexity' terminology to emphasize that our blowing snow scheme does not rely on a sophisticated surface snow scheme that explicitly account for densification effects associated with snow erosion (such as SNOWPACK in CRYOWRF for instance). Moreover, we want to stress that we consider a relatively simple one-moment treatment for the blowing snow water species (unlike a 2-moment treatment in CRYOWRF and Méso-NH) and that one scheme does not include an additional vertical discretization of the surface layer such as that in CRYOWRF and Méso-NH). We have rewritten the parameterization introduction paragraph as follows: We therefore follow an intermediate-complexity approach in the sense that the parameterisation does not require a very sophisticated snow scheme - such as SNOWPACK for CRY-OWRF for instance [6] - and does not include an additional discretization of the surface layer such as in Vionnet et al. [8]. Such as in MAR [2] and RACMO [3], a blowing snow flux is directly calculated between a fully parameterised saltation layer near the surface and the first model level at a few meters above the ground surface. However, the specific content of blowing snow particles in suspension  $q_h$  (in kg kg-1) is treated as an independent water variable in the model - unlike in MAR for instance - to properly distinguish the blowing snow contribution to precipitation and radiative effects from that of typical clouds.  $q_b$  is advected by the dynamical core and vertically transported by turbulent diffusion. However, we keep a one-moment treatment for the blowing snow water species and does not consider an additional prognostic estimation of the number of blowing snow particles [8, 6].

2. Line 115: What's the justification for the use of threshold friction velocity of 0.211 m/s? In Gallee (2001) u\*t0 is a variable (Eq. 3 in Gallee 2001) and subsequent implementations in RACMO and other models use this (although with some assumptions wrt snow dendricity etc). Using a constant u\*t0 would influence the quantity of blowing snow in the model. Where is this number coming from, please justify.

Thank you very much for pointing this mistake out. In fact, the equation was recopied from a report in which we did a sensitivity experiment with a fixed value of  $u_{*t0}$ . In the code, this variable is not constant and does depend on the surface drag coefficient (as in Gallée, Guyomarc'h, and Brun [2] and Amory

et al. [1]):
$$u_{*t0} = \frac{\log 2.688 - \log 1.625}{0.085} C_D^{0.5} \tag{1}$$

However; what is hidden in the  $\log 1.625$  is the fact that we assume that the snow grains have fixed dendricity and sphericity values d = s = 0.5, such as in Amory et al. [1]. The text introducing  $u_{*t}$  has been corrected accordingly.

- 3. Equation 3 seems incorrect, it must be  $0.08345u*^{1.27}$  (Pomeroy and Male 1992, Eq. 37). What exactly was implemented in the code? Thank you for pointing this mistake which has been corrected in the manuscript. The implementation in the code was however correct.
- 4. Line 204: Are the idealized simulations performed within the global model run or is it an offline run? And what is the 'oscillating behaviour' being talked about here?

These simulations were realized with a toy (offline) model of Eq. 11. The wording 'oscillating behaviour' was awkward, we meant that the proposed numerical scheme is numerically stable. The corresponding sentence was rewritten in that regard, and the caption of Fig. 1 was modified to specify that the curves shown correspond to a toy model of Eq. 11.

5. Line 216: Agree, but there is no description about the particle sizes considered in the study which is a critical parameter influencing the blowing snow flux. Please include that in the revised paper.

This is indeed an important aspect, especially for blowing snow sublimation. As a first approach, we assume a mono-disperse population of blowing snow particles whose radius is constant. This is a tuning parameter that can be controlled in the namelist file and whose default value is  $50~\mu m$ . Following a comment by another reviewer, we have also implemented the height-dependent blowing snow radius formulation of [5] but this option has not been fully evaluated yet. This is now better explained in the manuscript in Section 2.5.

6. Fig 3: Elevation changes due to snow deposition makes the acoustic tubes submerged and the flux might not be representative of the average flux at 1 m and 2m. Did you account for the elevation changes? For explanation, see Amory (2020) and Gadde and van de Berg (2024) Eq. 11.

Thank you for raising this important point which has also been noticed by the other referees. Indeed the lowermost FlowCapt sensor regularly gets partially buried – as illustrated in Figure ?? (see response to Referee 1 above) – and the accumulated snow height can be estimated thanks to a SR50 depth acoustic sensor. However, the SR50 was deployed in December 2012 at D17, and only few information about surface elevation is available in 2011, that is during the analysis period considered in the present study. In fact, the station the instruments are raised back manually to original heights at the beginning of each summer field campaign so the flux is likely subject to an underestimation especially in winter and spring. Unfortunately, no scaling correction can be properly applied

on D17 data. At D47, as the SR50 was operational throughout the 2011 year, we apply the same correction as in Amory et al. [1] to compute the flux vertically averaged along the wind-exposed part (h) of the sensor (of full height H):  $F_{b,corrected} = F_{b,measured} \times H/h$ . All the figures and tables have been modified accordingly. A new paragraph has also been added in Sect. 3.1.3 to explain the correction:

Throughout the year, the lowermost FlowCaptTM gets partially buried due to snow accumulation. At D47, a SR50 acoustic depth sensor monitored the surface elevation continuously between 2010 and 2012 showing that the wind-exposed part of the H=1 m high sensor was  $h\approx 0.6$  m in 2011. Building from Amory et al. [1], the measured flux has therefore been scaled at each time step by H/h to obtain the particle mass flux vertically averaged over the wind-exposed part of the sensor, consistently with the sensor calibration principle which implicitly assumes integration over its full exposed height H, requiring correction when only a fraction h is exposed. At D17, the SR50 sensor was deployed in December 2012, thus after the 2011 analysis period considered here. No correction can therefore be applied for this station which likely results in a underestimation of the flux magnitude. As the D17 instruments are raised back manually to original heights at the beginning of each summer field campaign, the underestimation is likely more important during the winter and spring season but this cannot be properly quantified.

7. Figure 5: Why was this not plotted for D17? Please include the figure in revised manuscript for consistency.

A panel for D17 has been added in the figure for consistency and it is now commented in the main text along with that on D47

8. Line 375 and Figure 5: Blowing snow flux has non-zero value at lower velocities when compared to the observations, this perhaps has to do with the assumption of constant threshold friction velocity of 0.211 m/s.

Please see our answer to your second comment about the constant  $u_{*t0}$  value. The fact that our model slightly overestimates the magnitude of the blowing snow flux at D47 for low wind speeds is indeed noticeable. Those conditions occur either far from snowfall event or during weak snowfall events. This overestimation can thus be attributed to the surface snow densification parameterization and/or to the LMDZ precipitation scheme that would generate an excess of fresh snow at the surface during such events. The text has been modified accordingly: .

At low wind speed - which generally corresponds to situations far from snowfall events or corresponding to weak snowfall events - the model tends to overestimate the flux which might be attributed to a too slow surface snow densification or excessive simulated snowfall by the LMDZ precipitation scheme leading to an excess in surface fresh snow. At high wind speed values, blowing snow flux observations show a more pronounced slope.

9. Line 487-488: Is the blowing snow variable defined at all the model levels

i.e. 95 model levels for the LAM run and 60 model levels for the bigger run? This seems like an overkill. Observations and meso-scale simulations are pretty consistent that the blowing snow phenomenon is mostly a lower boundary layer phenomenon. See Palm et al. 2017, RACMO results from Gadde and van de Berg (2024) (Fig. 5b). Gadde and van de Berg (2024) use only 16 grid points for the blowing snow model, with finer grid near the surface and results show good agreement with the observations without significant computational overhead. Please add in the discussion reason for not taking the standard approach of including the physics comparing it with your approach.

We completely agree with the referee that blowing snow is first and foremost a near surface process even though ground-based and satellite radar measurements have evidenced Antarctic blowing snow layers exceeding 1000 m (e.g., [7, 4]). The additional computational cost of blowing snow has two origins: the new physical parameterizations (in the physics of the model) and the advection of a new water species (in the dynamical core). Most of the additional cost in fact comes from the advection (not shown). We indeed could limit the treatment of blowing snow processes to the first layers above ground surface (stopping the vertical loops at a given model level for instance) but the overall computational gain would be very limited. Currently, there is no possibility in our dynamical solver DYNAMICO to constrain the transport of tracers over a subdomain. This might be some prospect development work but that goes well beyond the scope of the present paper. We have added in Sect. 3.1.1:

'It is worth mentioning that the additional computational cost of blowing snow mostly comes from the advection of a new water species in the dynamics rather than the treatment of the new parameterizations (surface snow erosion, turbulent transport, sedimentation and sublimation) in the physics part of the model. In the global configuration, this additional cost is about +4%.'

10. Add the computational cost of Blos vs No-Blos simulations. The additional computational cost of adding blowing snow in global runs is about 4%. This is now mentioned in the main manuscript.

**Minor comments**

- 1. Line 45 : climate global run's constrains  $\rightarrow$  global climate run's constraints Corrected.
- 2. Line 26: (- that we will hereafter combine into the single denomination of blowing snow for convenience -) too wordy rephrase with 'hereafter blowing snow'.

We have rephrased as follows: ''Hereafter, we will combine blowing and drifting snow into the single denomination of blowing snow for convenience.'

3. Line 113: Density terms need to be an exponent according to Gallee (2001)?? Equation number is also missing.

Equation number has been added. An exponential was indeed missing in the equation, thank you for noticing. This has been corrected.

- 4. Line 295: closet  $\rightarrow$  closest Corrected.
- 5. Line 345: resp.??? Changed to 'respectively'.
- 6. Line 423: tens of K? or few tenths of K? Corrected.
- 7. Line 510: While you mention that the code can be downloaded freely from the LMDZ website, it seems it is really not that straightforward. I tried to have a look at the blowing snow parameterisation, but could not figure out where to download the svn version that you used. If possible, please share the code/physics modules used in an easily accessible public repository for the benefit of the readers.

Thank you for raising this difficulty. This point was also raised by the Executive Editor. We now share the version of the model used to produce the results through a zenodo repository. Please see the answer to the Executive Editor comment for more details.

**References**

- [1] C. Amory et al. "Performance of MAR (v3.11) in simulating the drifting-snow climate and surface mass balance of Adélie Land, East Antarctica". In: Geoscientific Model Development 14.6 (2021), pp. 3487-3510. DOI: 10. 5194/gmd-14-3487-2021. URL: https://gmd.copernicus.org/articles/14/3487/2021/.
- [2] H. Gallée, G. Guyomarc'h, and E. Brun. "Impact of snow drift on the Antarctic Ice Sheet surface mass balance. Possible sensitivity to snow surface properties". In: *Boundary-Layer Meteorol* 99 (2001), pp. 1–19.
- [3] J. T. M. Lenaerts et al. "Modeling drifting snow in Antarctica with a regional climate model: 1. Methods and model evaluation". In: Journal of Geophysical Research: Atmospheres 117.D5 (2012). D05108. DOI: 10.1029/ 2011JD016145.
- [4] S. P. Palm et al. "Blowing snow sublimation and transport over Antarctica from 11 years of CALIPSO observations". In: *The Cryosphere* 11.6 (2017), pp. 2555-2569. DOI: 10.5194/tc-11-2555-2017. URL: https://tc.copernicus.org/articles/11/2555/2017/.

- [5] Manuel Saigger et al. "A Drifting and Blowing Snow Scheme in the Weather Research and Forecasting Model". In: *Journal of Advances in Modeling Earth Systems* 16.6 (2024), e2023MS004007. DOI: https://doi.org/10.1029/2023MS004007.
- [6] V. Sharma, F. Gerber, and M. Lehning. "Introducing CRYOWRF v1.0: multiscale atmospheric flow simulations with advanced snow cover modelling". In: Geoscientific Model Development 16.2 (2023), pp. 719-749. DOI: 10.5194/gmd-16-719-2023. URL: https://gmd.copernicus.org/articles/16/719/2023/.
- [7] Etienne Vignon et al. "Gravity Wave Excitation during the Coastal Transition of an Extreme Katabatic Flow in Antarctica". In: *Journal of the Atmospheric Sciences* 77.4 (2020), pp. 1295–1312. DOI: 10.1175/JAS-D-19-0264.1.
- [8] V. Vionnet et al. "Simulation of wind-induced snow transport and sublimation in alpine terrain using a fully coupled snowpack/atmosphere model".
  In: The Cryosphere 8.2 (2014), pp. 395-415. DOI: 10.5194/tc-8-395-2014.
  URL: https://tc.copernicus.org/articles/8/395/2014/.

---

## Author Comment (AC4)

**Revision of**

**'Intermediate-complexity Parameterisation of Blowing Snow in the ICOLMDZ AGCM: development and first applications in Antarctica'**

Etienne Vignon, Nicolas Chiabrando et al.

October 31, 2025

This document contains the response to a review of 'Intermediate-complexity Parameterisation of Blowing Snow in the ICOLMDZ AGCM: development and first applications in Antarctica' submitted to EGUSPHERE for possible publication in Geoscientific Model Development. Comments from the Editor are in black and answers are in blue. Paragraphs that have been added or modified during the revision process are copied in purple.

**1 Comment from Juan A. Añel, Executive Editor**

Dear authors,

Unfortunately, after checking your manuscript, it has come to our attention that it does not comply with our "Code and Data Policy".

https://www.geoscientific-model-development.net/policies/code\_and\_data\_policy.html

You have archived your code on servers not suitable for scientific publication (e.g., a svn in jussieu.fr and GitLab sites). You must store all the assets necessary to replicate your manuscript in a suitable repository, from the ones listed in our policy. Also, you have not published the output data from your simulations, and you must do it. Therefore, the current situation with your manuscript is irregular.

Please, publish your code and data in one of the appropriate repositories and reply to this comment with the relevant information (link and a permanent identifier for it (e.g. DOI)) as soon as possible, as we can not accept manuscripts in Discussions that do not comply with our policy.

Also, remember to include a modified Code and Data Availability sections in a potentially reviewed manuscript, containing the information of the new repositories.

I must note that if you do not fix this problem, we can not continue with the peer-review process or accept your manuscript for publication in our journal. Dear Executive Editor, thank you for raising this issue of non-compliance with the journal policy regarding the storage and share of our code and simulation materials. The exact code version and input files used to run the simulations are now shared on zenodo along with the simulation output and codes to reproduce the figures (10.5281/zenodo.17493828). The Code and Data availability section has been rewritten as follows:

The current version of LMDZ and DYNAMICO are available from the project websites http://www.lmd.jussieu.fr/~/pub and https://gitlab.in2p3.fr/ipsl/projets/dynamico/dynamico under CeCILL licence. The exact version of the model used to produce the results is archived on repository under DOI along with input data and scripts to run the model and produce the plots for all the simulations presented in this paper [1]. Boundary condition files for the limited-area simulations have been built using the data-rigueur software, freely distributed at this site: https://gitlab.in2p3.fr/ipsl/projets/awaca/models/data-rigueur. The scripts used for the SMB analysis and evaluation using SMB observational data are distributed here: https://gitlab.in2p3.fr/ipsl/projets/awaca/modelobs/smb-transects-antarctica-git/.

**References**

[1] Etienne VIGNON. Material to reproduce the results of ICOLMDZ simulations with blowing snow, for the revision of Vignon et al. 2025. Zenodo, Oct. 2025. DOI: 10.5281/zenodo.17493828. URL: https://doi.org/10.5281/zenodo.17493828.

---

## Author Response (AR2)

This document contains the response to the minor review of `Intermediate-complexity Parameterisation of Blowing Snow in the ICOLMDZ AGCM: development and first applications in Antarctica' submitted to EGUSPHERE for possible publication in Geoscientific Model Development.

Comments from the Reviewer are in black and responses are in blue.

Review of the revised version of egusphere-2025-2871: 'Intermediate-complexity Parameterisation of Blowing Snow in the ICOLMDZ AGCM: development and first applications in Antarctica' by Vignon et al.

I thank the authors for thoroughly addressing my comments and answering my questions. I still think that the vertical extrapolation of the FlowCapt measurements is highly uncertain and the associated uncertainties should be stated more clearly (see comment 2 below). Additionally, I have some other minor comments but recommend publication after they are addressed. The line numbers below refer to the revised manuscript (not the track-changes version).

We sincerely thank the Reviewer for the second thorough review of our manuscript. Please find below our detailed responses to each comment.

(1) l. 340: "between 0 and 1 m a.g.l. or between 1 and 2 m a.g.l.": Here (and in other instances, e.g., caption of Fig. 3), I suggest to add the word "approximately" or "roughly" as the values change with time when the sensor gets more and more buried in the snow.

We agree and the word 'approximately' has been added everywhere necessary in the manusript.

(2) l. 346-347: "which results in an overall exponential decay of the flux with increasing height (Mann et al., 2000; Gordon et al., 2009; Sigmund et al., 2025)": This statement does not apply to the suspension layer and should be modified. The cited studies suggest an exponential mass flux profile in the saltation layer only. In the suspension layer, however, the blowing snow concentration (and similarly the mass flux) is expected to be close to a power-law profile of height (the flux decreases less strongly with height, compared to the exponential profile of the saltation layer). This difference between the saltation layer and suspension layer is supported by most mass flux profiles measured by Nishimura and Nemoto (2005) (their Figure 9). As the lower FlowCapt sensor averages over both the saltation layer and a part of the suspension layer, it is difficult to predict which profile function would be most suitable for extrapolation of the FlowCapt measurements. This uncertainty should be mentioned.

Thank you for this comment, we totally agree. Following your suggestions, the text has been modified as follows :
'One possibility for the D47 site is to compute a mean value over the first model layer depth after a vertical extrapolation of the flux from the measurements of the two superimposed 2G-FlowCaptTM The vertical profile of the particle mass flux follows an exponential decay in the saltation layer (Martin et al. 2017, Melo et al. 2024) which results in an overall exponential decay of the flux with increasing height (Mann et al. 2000, Gordon et al. 2009, Sigmund et al. 2025) In the suspension layer however, the blowing snow concentration and the blowing snow mass flux are expected to be close to a power-law profile of height (Nishimura & Nino 2005). As the lower FlowCapt sensor averages over both the saltation layer and a part of the suspension layer, it is difficult to predict which profile function would be most suitable for extrapolation of the FlowCapt measurements. Although uncertain, an exponential extrapolation of the form Fb(z)=Fb0 exp(-z/Hb) is used here as a first approach, ……'

(3) l. 355: "local flux measurements at 1 and 2 m": The wording is misleading as it sounds like point measurements at heights of 1 and 2 m. Maybe write: "local flux measurements of both sensors in the lowest 2 m".

Corrected.

(4) caption of Fig. 3: "measurements between 0 and 1 m": To avoid misunderstandings, consider writing: „measurements between 0 and ~1 m after correcting for the partial burial of the sensor"

Corrected.

(5) l. 413: The underestimated increase of the mass flux with wind speed in the model might also be due to simplifications in the saltation model of Pomeroy (1989), affecting the predicted relationship between the blowing snow concentration at the top of the saltation layer and friction velocity (Eq. 7).

We have added in the text :
'The underestimated increase of the mass flux with wind speed might also be explained by the overly simple saltation model of \citet{Pomeroy_1989} considered here, which can affect the predicted relationship between the blowing snow concentration at the top of the saltation layer and the friction velocity.'

(6) l. 501: „Their amplitude is also fairly well reproduced": Given the considerable uncertainties, especially due to the vertical extrapolation of the measurements, I suggest to write: „The order of magnitude of this flux is also fairly well reproduced."

Corrected.

Technical corrections:
(7) l. 91: "studies complex terrains areas" should probably be "studies in complex terrain areas"
Corrected

(8) l. 95: "at a regional and continental scales" should be "at regional and continental scales."
Corrected

(9) l. 108: "does" should be "do".
Corrected

(10) l. 125: "ans" should be "and".
Corrected

(11) l. 311: "a underestimation" should be "an underestimation".
Corrected

(12) l. 311: "0.072 g m-2 s-1" should probably be "0.140 g m-2 s-1" as in the previous sentence, where it was changed during revision.
Corrected

(13) l. 570: DOI is missing

Corrected

References (apart from those listed in the manuscript):

Nishimura, K., & Nemoto, M. (2005). Blowing snow at Mizuho station, Antarctica. Phil. Trans. R. Soc. A. 363, 1647–1662. https://doi.org/10.1098/rsta.2005.1599